# A bidentate Polycomb Repressive-Deubiquitinase complex is required for efficient activity on nucleosomes

Martina Foglizzo [1], Adam J. Middleton [1], Abigail E. Burgess [1], Jennifer M. Crowther [2], Renwick C.J. Dobson [2,3], James M. Murphy [4,5], Catherine L. Day [1] & Peter D. Mace [1]

Attachment of ubiquitin to lysine 119 of Histone 2A (H2AK119Ub) is an epigenetic mark characteristic of repressed developmental genes, which is removed by the Polycomb Repressive-Deubiquitinase (PR-DUB) complex. Here we report the crystal structure of the *Drosophila* PR-DUB, revealing that the deubiquitinase Calypso and its activating partner ASX form a 2:2 complex. The bidentate Calypso–ASX complex is generated by dimerisation of two activated Calypso proteins through their coiled-coil regions. Disrupting the Calypso dimer interface does not affect inherent catalytic activity, but inhibits removal of H2AK119Ub as a consequence of impaired recruitment to nucleosomes. Mutating the equivalent surface on the human counterpart, BAP1, also compromises activity on nucleosomes. Together, this suggests that high local concentrations drive assembly of bidentate PR-DUB complexes on chromatin—providing a mechanistic basis for enhanced PR-DUB activity at specific genomic foci, and the impact of distinct classes of PR-DUB mutations in tumorigenesis.

[1] Biochemistry Department, School of Biomedical Sciences, University of Otago, P.O. Box 56, 710 Cumberland St., Dunedin 9054, New Zealand. [2] Biomolecular Interaction Centre, School of Biological Sciences, University of Canterbury, Christchurch 8140, New Zealand. [3] Bio21 Molecular Science and Biotechnology Institute, Department of Biochemistry and Molecular Biology, University of Melbourne, Parkville, VIC 3010, Australia. [4] The Walter and Eliza Hall Institute of Medical Research, Parkville, VIC 3052, Australia. [5] Department of Medical Biology, University of Melbourne, Parkville, VIC 3052, Australia. Correspondence and requests for materials should be addressed to P.D.M. (email: peter.mace@otago.ac.nz)

Epigenetic control of transcription is relevant to all eukaryotic cell biology and underlies many human diseases. Covalent modifications to histone proteins (also known as histone 'marks') are one of the most well-established mechanisms of epigenetic regulation, and are vital to packaging of DNA, recruiting transcription factors, and ultimately gene transcription[1]. Consequently, the enzymes that attach and remove histone marks are among the most frequently dysregulated genes in cancer[2,3]. Of particular relevance is attachment of ubiquitin to Histone protein 2A (H2A) on lysine residue 119 (H2AK119Ub) in humans. H2AK119Ub is one of the most prevalent histone marks, and is estimated to occur on 5–15% of histones genome-wide[4]. In general, H2AK119Ub is characteristic of repressed developmental genes, and is also tightly linked to levels of other histone marks such as lysine methylation and acetylation[5].

H2AK119Ub is attached to histones by the Really Interesting New Gene (RING) ubiquitin E3-ligase Polycomb Repressive Complex 1 (PRC1), and removed by the Polycomb Repressive-Deubiquitinase (PR-DUB) complex[6]. The PR-DUB was first characterised in *Drosophila melanogaster*, where it consists of the deubiquitinase protein Calypso and a binding partner Additional Sex Combs (ASX)[7]. The human PR-DUB complex consists of the deubiquitinase Breast Cancer type 1 susceptibility protein (BRCA1)-associated protein-1 (BAP1) and one of three regulatory ASX-Like proteins (ASXL1–3)[7–9]. Mutations in BAP1 are responsible for BAP1 tumour predisposition syndrome (BAP1-TPDS; OMIM number: 614327), which drives development of mesothelioma, melanoma, and other neoplasms; sporadic mutations in BAP1 and ASXL1–3 also frequently occur across various cancer types[10–13]. BAP1-TPDS is highly penetrant, with ~85% of heterozygous carriers diagnosed with cancer[14]. Data from both mice and humans demonstrate that BAP1 mutations lead to significantly increased risk of malignancy from environmental carcinogens[10]. For instance, heterozygous mice with one wild type and one null BAP1 allele develop mesothelioma at much lower levels of exposure to asbestos than wild-type animals. Moreover, BAP1 mutations are found in ~15% of clear-cell renal-cell carcinomas (CCRCC), and patients bearing such mutations have a particularly poor prognosis relative to other common molecular subtypes of CCRCC[15].

BAP1 and Calypso each contain a catalytic ubiquitin C-terminal hydrolase (UCH) domain, and a C-terminal UCH37-like domain (ULD) (Fig. 1a and Supplementary Fig. 1a) that recruits the deubiquitinase adaptor (Deubad) domain in ASX, or ASXL1–3 (refs. [7–9]). The overall domain architecture is shared between the human and *Drosophila* PR-DUB components, with the exception of a large insertion (~380 amino acids) in the ULD domain of BAP1 (Fig. 1a and Supplementary Fig. 1). Finally, BAP1 and Calypso share a positively charged tail at their C-termini (Fig. 1a and Supplementary Fig. 1) that has recently been shown to enhance recruitment of the human PR-DUB complex to substrate nucleosomes[9]. No structures of any PR-DUB complex have been reported, but inferences have been extended from the human ortholog UCH-L5, which also contains an N-terminal UCH domain and a C-terminal ULD domain. Recent studies have shown that substrate binding and catalysis by UCH-L5 is promoted by the Deubad of the proteasome-associated adapter Rpn13, or inhibited by the Deubad of the chromatin remodelling complex subunit INO80G, with the Deubad from each binding partner stabilising different conformations of the UCH-L5 ULD domain[16,17]. While the UCH-L5–Rpn13 complex has provided a template for PR-DUB catalysis and modelling of some cancer-derived mutations[8,9,18], many outstanding questions remain unanswered. For instance: it is not clear what the active oligomeric state of the PR-DUB is; how the PR-DUB is recruited to nucleosomes; or why BAP1 exhibits haploinsufficiency[19,20],

where a single functional copy of BAP1 is not sufficient to protect from the effects of environmental carcinogens. More broadly it is unclear why ASXL1 and ASXL2 truncation mutations can cause either loss-, or gain-of-function;[21,22] or why cancers caused by BAP1 do not overlap with those caused by ASXL1/2 loss[23].

With the goal of understanding how PR-DUB activity is coordinated, we have solved the crystal structure of the *Drosophila* Calypso–ASX complex to a resolution of 3.5 Å. Using size-exclusion chromatography coupled to multi-angle laser light scattering (SEC-MALLS), analytical ultracentrifugation (AUC), small-angle X-ray scattering (SAXS), and chemical crosslinking we demonstrate that Calypso–ASX forms a higher-order complex in solution containing two Calypso and two ASX molecules. Oligomer assembly is mediated by a conserved patch on the surface of the coiled-coil of Calypso, and disruption of this interface impairs recruitment of the PR-DUB to nucleosomes and removal of H2AK119Ub marks. Together, this study suggests a mechanism where a bidentate PR-DUB complex with two deubiquitinases is required for full recruitment and activity on nucleosomes. Bidentate complex assembly and maximum activity is most likely to occur at high local PR-DUB concentrations that arise following targeting to specific regions of the genome.

## Results

**Structure of the *Drosophila* PR-DUB.** To understand the determinants of PR-DUB catalytic activity, we characterised the complex between Calypso and the Deubad domain of ASX. We focused on Calypso because it contains the major structural elements required for activity of the PR-DUB, but lacks the large insertion present in the ULD domain of BAP1 that is predicted to be disordered. Following co-expression in *Escherichia coli*, the Calypso–ASX complex was crystallised and after extensive optimisation the structure was solved to a resolution of 3.5 Å (Supplementary Table 1). Two copies each of Calypso and ASX are present in the asymmetric unit (Fig. 1b). These are highly similar, except one copy of the Deubad domain is better defined, owing to more extensive crystal packing.

In each of the two copies of the Calypso–ASX complex the globular Calypso UCH domain is linked to the C-terminal ULD region by a coiled-coil hairpin. The Deubad domain of ASX packs at the hinge between the coiled-coil and ULD helix of Calypso, restraining the ULD in a pose protruding away from the active site (Fig. 1b). The 1:1 topology is similar to that observed for the related deubiquitinase UCH-L5 bound to its proteasomal activator Rpn13 (refs. [16,17]) (Supplementary Fig. 2, left panel). Superposition of the trapped UCH-L5~Ub–Rpn13 complex (PDB code: 4UEL)[16] onto the structure shows that the arrangement of Calypso–ASX builds a composite binding site competent to accommodate ubiquitin (Fig. 1c). The Deubad of ASX in the crystal structure is markedly shorter in its C-terminal portion than the Rpn13 Deubad, a region that is also relatively divergent between Rpn13 and ASX-type Deubad sequences (Supplementary Fig. 1d). Whereas Rpn13 consists of seven helices that wrap around the ULD sequence of UCH-L5, only four are defined for ASX. It is possible that in situ proteolysis used during crystal growth may have removed residues 310–340 of the ASX Deubad, or that the remaining residues of the ASX Deubad adopt alternate conformations in the crystal. Either scenario would suggest a degree of conformational flexibility in the C-terminal region of ASX Deubad.

**Characteristics of missense mutations in the human PR-DUB.** Putative driver mutations in PR-DUB components frequently occur across various cancer types[10,24–28]. To understand common missense variants, we mapped mutations in the human Calypso

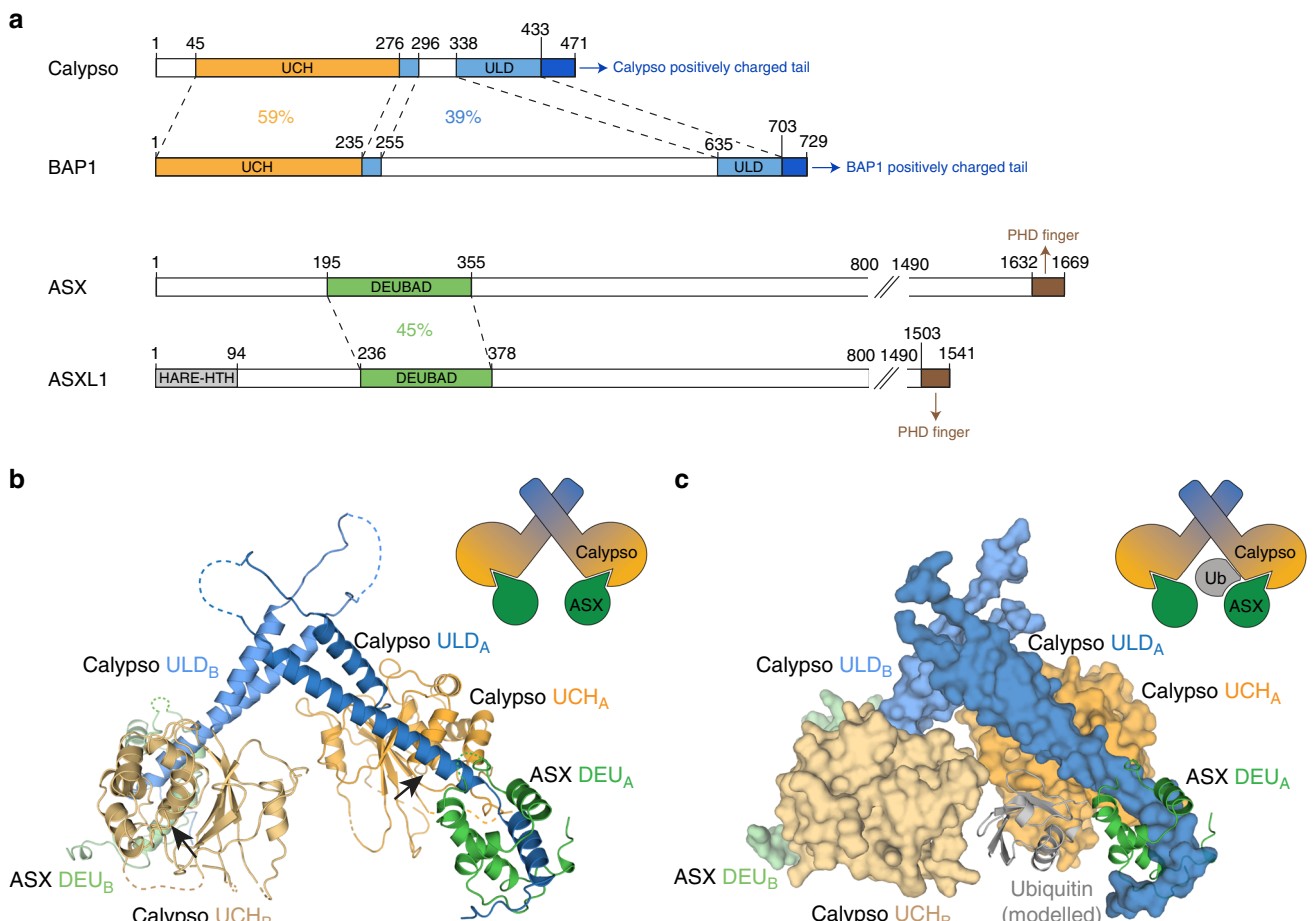

**Fig. 1** Crystal structure of the *Drosophila* PR-DUB complex. **a** Schematic representing domain structure of human and *Drosophila* PR-DUB components. Sequence identity between the UCH and ULD domains of Calypso and BAP1, and between the Deubad domains of ASX and ASXL1 is indicated. FL, full-length; UCH, ubiquitin C-terminal hydrolase; ULD, UCH37-like domain; ASX, additional sex combs; ASXL1, ASX-like 1; PHD, plant homeodomain (see Supplementary Fig. 1a). **b** Structure of the Calypso–ASX complex (PDB code: 6CGA). The UCH and ULD domains of Calypso are coloured orange and blue, respectively. The Deubad domain of ASX is coloured green. Black arrows indicate the position of the active site cysteine residue in the respective UCH domains. Deubad (DEU), deubiquitinase adaptor domain (see Supplementary Table 1). **c** A model of the Calypso–Ub–ASX complex (based on PDB 4UEL; ref. [16]). The Calypso–ASX structure is shown in surface representation and colour coded as in **b**, while the modelled ubiquitin and the ASX Deubad domain in chain A are shown as cartoon and coloured grey and green respectively. Ub, ubiquitin

ortholog, BAP1, from all tumour samples available in cBioPortal[25,26], onto the Calypso–ASX structure (Fig. 2a and Supplementary Table 2). Mutations are clearly enriched in key functional regions of the catalytic domain, with multiple occurrences within the active site triad (Cys91, His169, and Asp184), the ubiquitin binding cleft (Fig. 2a and Supplementary Table 2), and the crossover-loop that spans the active site of UCH DUBs and potentially impacts substrate selectivity. Notably Gly185 and Arg227, located in the active site cleft and at the ubiquitin interface, respectively, have the highest frequency of mutations. A range of missense mutations pepper the ~380 amino acid insertion of BAP1, which is important for assembly of large multi-protein complexes[29–33]. Some mutations also affect the C-terminal tail of BAP1, consistent with the importance of this region in multiple aspects of BAP1 function, including its auto-deubiquitination, cellular localisation, and nucleosome-binding properties[9,34].

The *Drosophila* ASX protein has three human orthologs (ASXL1–3), which also possess an N-terminal Deubad and a C-terminal plant homeodomain (PHD). Truncating mutations in ASXL1 and ASXL2 that remove the PHD, but retain the Deubad, have been linked to both myeloid and myelomonocytic

leukaemias[21,22,35], as well as Bohring–Opitz syndrome[36]. In recently published work, Sahtoe et al.[9] and Peng et al.[18] investigated a conserved 'NEF' region in the Deubad domain of ASX-like proteins (residues Asn283–Glu284–Phe285 in ASX) (Supplementary Fig. 1d)[9,18]. The crystal structure reveals that the NEF-motif sits directly adjacent to ubiquitin in a putative substrate complex (Fig. 2b and Supplementary Fig. 3a). We noted recurring missense mutations relevant to the 'NEF' loop in cBioPortal: Glu330 in ASXL2 is mutated in breast, bladder, and urothelial tumours; His315 in ASXL1 in colon, colorectal, lung cancer, and melanoma patient samples (Supplementary Table 3). ASXL2 Glu330 is identical in ASX (i.e. Glu284), while the ASX equivalent to ASXL1 His315 is a positive arginine residue (Arg288; Fig. 2b and Supplementary Fig. 3a). To ascertain the functional impact of the observed mutations at these positions, we introduced E284K, E284Q, and R288N mutations in and around the ASX NEF-motif (Supplementary Table 3), and prepared Calypso–ASX complexes. Consistent with the Calypso–ASX structure, all mutant complexes lost the ability to hydrolyse Ubiquitin-AMC or a model ubiquitin-peptide substrate, and had reduced capacity to bind ubiquitin in pulldown assays (Fig. 2c and Supplementary Fig. 3b, c).

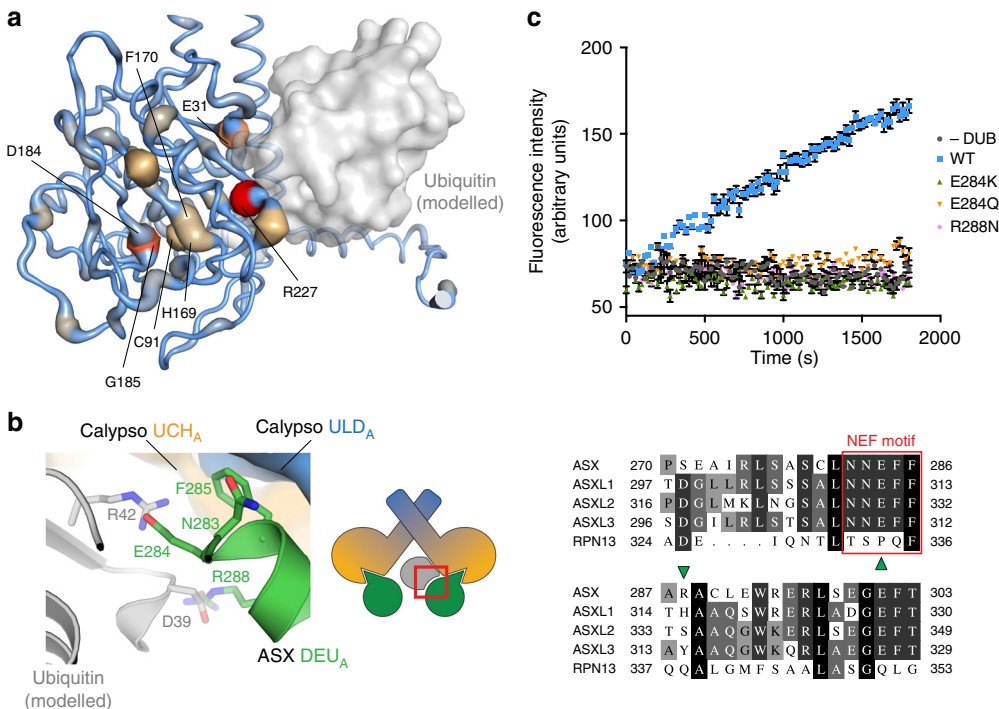

**Fig. 2** Characteristics of missense mutations in the human PR-DUB. **a** Model of the Calypso-Ub–ASX complex generated as described in Fig. 1c, with tumour-derived missense mutations in BAP1 mapped onto the Calypso UCH domain. Residues mutated with low, medium, or high frequency are coloured blue, light orange, or red, respectively. Amino acids located in the active site cleft or at the ubiquitin interface are indicated (numbering refers to the BAP1 sequence; see Supplementary Table 2). The Deubad domain of ASX has been removed for clarity. **b** Close-up view showing the residues in the NEF-motif of ASX, including Arg288, and the putative interacting amino acids in the modelled ubiquitin (left). Multiple sequence alignment of the residues surrounding the NEF region in ASX, ASXL1, ASXL2, ASXL3, and of the corresponding residues in Rpn13 (right). The NEF-motif is highlighted with a red box; green arrows indicate the residues targeted for mutations. (see Supplementary Fig. 3a and Supplementary Table 3). **c** Ubiquitin-AMC hydrolysis assays comparing the activity of wild type and mutant Calypso–ASX complexes. Error bars indicate range of measurement ($n = 2$ independent experiments). DUB, deubiquitinating enzyme; WT, wild type (see Supplementary Fig. 3b, c)

Together these results illustrate the structural impact of missense variants in both catalytic and non-catalytic PR-DUB components that are enriched in tumours, and show that the crystal structure of Calypso–ASX is a promising model for predicting the effects of mutations in the human BAP1–ASXL1/2 complexes.

**The PR-DUB forms a bidentate complex**. A recent study of the human PR-DUB has reported a 2:1 complex, containing two BAP1 molecules for every ASX-like molecule[9,37]. In order to determine the stoichiometry of Calypso–ASX we first analysed the complex using SEC-MALLS, which showed a molecular weight of 110 kDa (Fig. 3a). This mass agrees with a complex consisting of two Calypso and two ASX molecules, which would have a theoretical mass of 109 kDa. To validate the 2:2 complex using an independent method we used AUC. Analysis of sedimentation velocity experiments using the $c(s)$ method showed a homogeneous population of the 2:2 complex at a concentration of 40 µM, with an $s_{(20,w)}$ of 5.78 S (Fig. 3b and Supplementary Fig. 4a, c). Sedimentation equilibrium experiments performed at the same concentration showed a molecular weight of 112.9 kDa ($\pm 0.3$ kDa), which is again consistent with a 2:2 Calypso–ASX complex (Supplementary Fig. 4b). At decreasing protein concentrations (1–18 µM), there is a shift in the sedimentation coefficient to smaller values (Fig. 3c), which is consistent with mass-action dissociation of the 2:2 Calypso–ASX oligomer (see Supplementary Fig. 4c for van Holde–Weischet analysis).

A dissociation constant in the µM range for the 2:2 complex suggests that the bidentate PR-DUB is unlikely to be constitutive. In contrast, 1:1 interactions between deubiquitinase and ASX-like proteins are 100–1000-fold tighter ($K_D$ ~6–18 nM)[16,17]. Thus, there appears to be two layers of PR-DUB assembly: a tight interaction between a deubiquitinase and an ASX-like protein, and a more transient higher-order complex that could represent an additional mechanism for regulating PR-DUB activity.

**PR-DUB oligomerisation via the coiled-coil hairpin**. To dissect higher-order oligomerisation of the PR-DUB we analysed various interaction surfaces within crystal contacts of the Calypso–ASX complex. Three possible interfaces capable of building a 2:2 oligomer from the Calypso–ASX unit were identified by PISA (Protein, Interfaces, Structures and Assemblies)[38] (Supplementary Fig. 5a, top row, and Supplementary Table 4; interfaces #3, #4, and #5). We designed mutations at each of the three interfaces, and tested their oligomerisation using analytical SEC. Mutation of the interface formed by the coiled-coil hairpin of Calypso clearly shifted the peak of Calypso(L340A)–ASX, whereas Calypso–ASX complexes bearing mutations at all other crystallographic interfaces eluted in an identical position to the wild-type complex (Fig. 4a). Having established that oligomerisation of Calypso–ASX occurs via the coiled-coil hairpin in Calypso, we targeted additional residues at this interface. We introduced M288R and N292R mutations (Supplementary Fig. 5a, top row; interface #5) and prepared Calypso–ASX complexes. Wild type and mutant proteins were then tested in

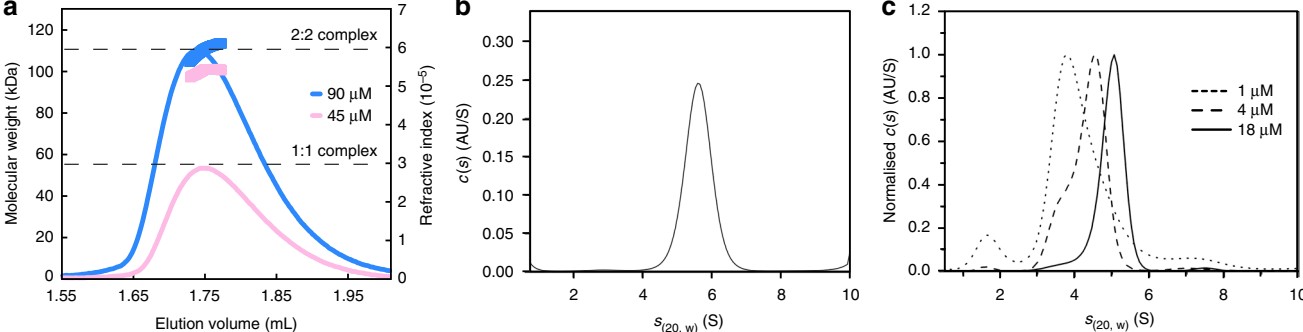

**Fig. 3** The *Drosophila* PR-DUB forms a bidentate complex. **a** SEC-MALLS analysis of Calypso–ASX. Elution profiles of the purified Calypso–ASX complex at 90 and 45 μM monitored by differential refractive indices (lines); the weight averaged molecular masses (Mw) are shown with symbols. Dashed lines correspond to the size of the 1:1 (55 kDa, bottom) and 2:2 (110 kDa, top) Calypso–ASX complex. **b** A $c(s)$ distribution of Calypso–ASX from sedimentation velocity AUC at 40 μM. The RMSD for the fit was 0.005 and the Runs-Test Z score was 2.25. The frictional ratio was floated and converged to a value of 1.00 (see Supplementary Fig. 4a, c). **c** $c(s)$ distributions of Calypso–ASX at 1, 4, and 18 μM. The residuals for each fit were randomly distributed with RMSD values of 0.008, 0.01, and 0.006 respectively (see Supplementary Fig. 4c)

analytical SEC experiments. Consistent with our previous data, the elution profiles of Calypso(M288R)–ASX and Calypso (N292R)–ASX were considerably shifted compared to the wild-type complex (Fig. 4b).

To characterise the contribution of each of these residues to Calypso–ASX oligomerisation, we performed SEC-MALLS experiments. Mutation of Leu340 and Asn292 to alanine and arginine residues respectively resulted in the formation of a 1:1 PR-DUB complex, with a molecular weight of 60 kDa (Fig. 4c). The Calypso(M288R)–ASX mutant exhibited an intermediate oligomeric state, with a molecular weight of 76 kDa (Fig. 4c). These results clearly indicate that Leu340 and Asn292 play a major role in Calypso dimerisation, and that mutation of Met288 also destabilises interactions at the 2:2 PR-DUB dimer interface.

The arrangement mediated by hydrophobic interactions between Leu340 and Met288 from the two Calypso coiled-coil hairpins (Supplementary Fig. 5) creates an unusual elongated 2:2 complex, which we investigated in solution using SEC-SAXS. Guinier analyses showed that the Calypso–ASX complex was monodisperse and free of aggregation, with a calculated $D_{max}$ of 125 Å (Fig. 4d, Supplementary Fig. 5f, and Supplementary Table 5). This was consistent with a 2:2 Calypso–ASX complex being present in solution, in line with SEC-MALLS and AUC (Fig. 3). To compare experimental and theoretical scattering profiles, we modelled and energy minimised several loops that were not defined by electron density in the crystal structure. Having generated a complete model (Supplementary Fig. 5g), scattering profiles were an excellent fit for the oligomers formed through either the Calypso UCH and Calypso coiled-coil hairpin (Supplementary Fig. 5a, bottom row; interfaces #3 and #5) whereas interface #4 led to higher Chi values and poorer fits (Supplementary Fig. 5a, bottom row).

Finally, we employed chemical crosslinking to interrogate the higher-order Calypso–ASX complex. Treatment with increasing concentrations of bis(sulfosuccinimidyl)suberate (BS³) (Fig. 4e), or with fixed concentrations of BS³ over increasing time periods (Supplementary Fig. 5h), led to crosslinked complexes that could be resolved by sodium dodecyl sulphate-polyacrylamide gel electrophoresis (SDS-PAGE). The first and most prominent complex migrated at ~55 kDa, consistent with a covalent 1:1 Calypso–ASX complex. A second tight group of crosslinked bands migrated with a molecular weight of approximately 150 kDa. Performing equivalent crosslinking experiments with Calypso(L340A)–ASX gave rise to a crosslinked band of ~55 kDa, but no higher molecular weight species (Fig. 4e and Supplementary Fig. 5h). Although SDS-PAGE mobility was

slightly aberrant relative to a theoretical mass of 109 kDa for the 2:2 complex, the clearest explanation for this profile is that the high molecular weight species contain Calypso–Calypso cross-links. These crosslinks are specific to wild-type Calypso–ASX rather than the L340A protein, and are consistent with L340A disrupting the Calypso–Calypso coiled-coil interface, but not disrupting the ability of Calypso to bind ASX.

Together, orthogonal biophysical and biochemical methods show that the bidentate PR-DUB complex is formed through an interface on the Calypso coiled-coil hairpin—in effect a Calypso dimer with each molecule independently able to bind one ASX Deubad domain.

**Bidentate complex assembly promotes activity on nucleosomes.** The arrangement of a 2:2 Calypso–ASX complex bridged by the Calypso coiled-coil creates an elongated structure with a large cleft separating the two UCH domains and the Deubad of ASX. To test whether mutations in the Calypso coiled-coil hairpin affect the intrinsic ubiquitin hydrolase activity of the PR-DUB, we first tested wild type and L340A Calypso–ASX proteins against a model ubiquitin-peptide substrate. Both complexes could hydrolyse the model substrate at comparable rates (Supplementary Fig. 3b). We then used Ubiquitin-AMC cleavage assays to determine the Michaelis–Menten constant for both wild type and L340A proteins. There was no significant difference in $K_M$ between the constructs (Fig. 5a), showing that mutation at Leu340 did not affect the correct folding of the complex or its catalytic activity.

Having established that disrupting the 2:2 Calypso–ASX complex into individual Calypso–ASX units does not affect its intrinsic catalytic activity, we next sought to test whether bidentate complex assembly was necessary for activity on its primary biological substrate, H2AK119Ub nucleosomes. We generated recombinant nucleosomes by co-expression in *E. coli*, and used the PRC1 E3-ligase pair Ring1b-Bmi1 to specifically ubiquitinate Histone 2A[39]. In order to visualise and quantitate assays we employed N-terminal 5-iodoacetamidofluorescein (5-IAF)-labelled ubiquitin. While wild-type Calypso–ASX robustly deubiquitinated H2AK119Ub, the activity of the Calypso (L340A)–ASX complex was severely attenuated (Fig. 5b). Performing equivalent experiments with the Calypso(M288R)–ASX and Calypso(N292R)–ASX proteins also resulted in decreased activity (Fig. 5b). Thus, bidentate complex assembly is crucial for the ability of the PR-DUB complex to remove H2AK119Ub from nucleosomes, even though it does not impair intrinsic catalytic activity.

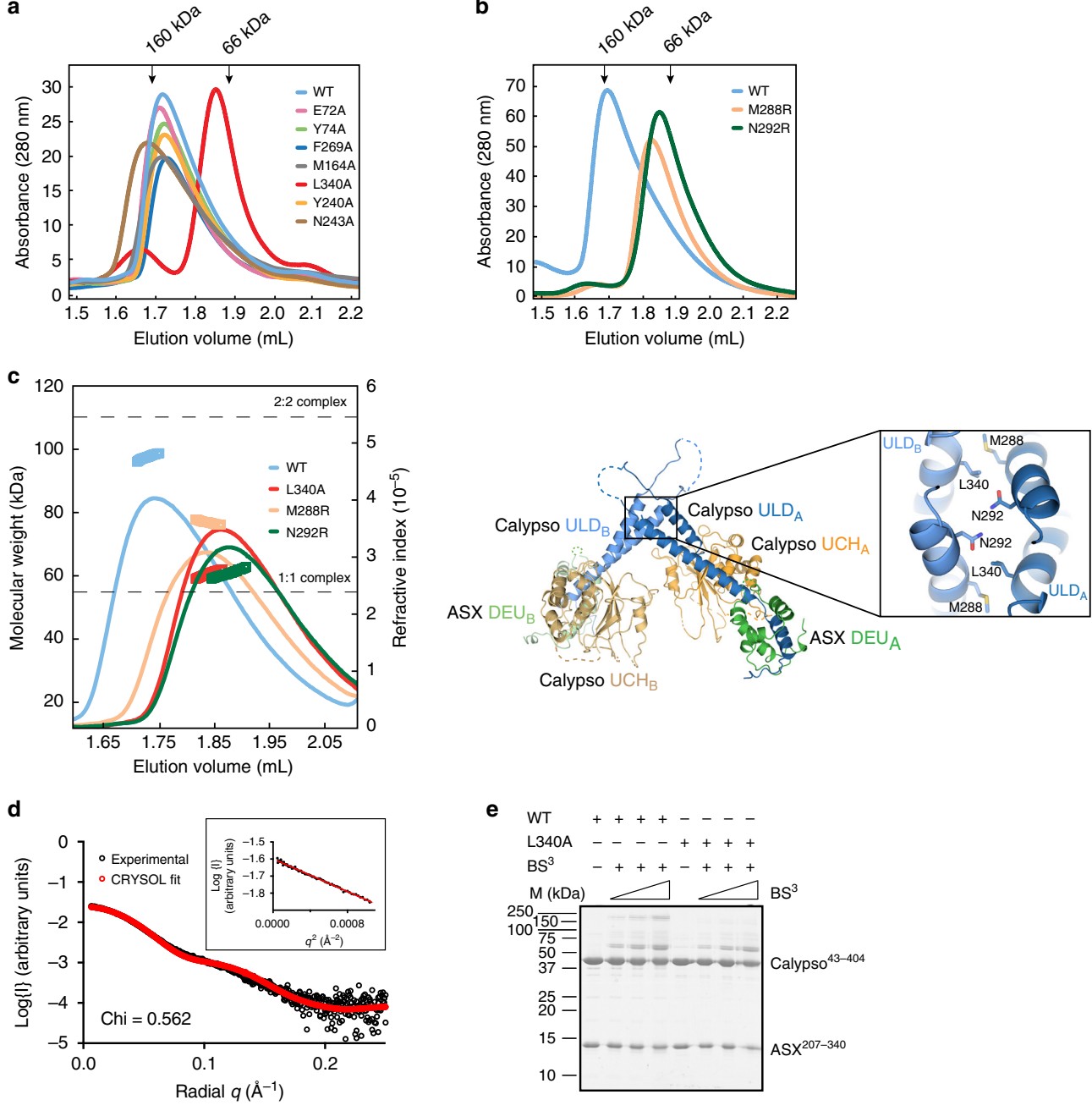

**Fig. 4** The Calypso coiled-coil hairpin mediates PR-DUB oligomerisation. **a** Analytical size-exclusion chromatography analysis of wild type and mutant Calypso–ASX complexes. For all runs, elution profiles at ~30–50 μM were recorded by monitoring absorbance at 280 nm. Black arrows indicate the elution profiles of protein standards. **b** Analytical size-exclusion chromatography analysis of wild type, M288R and N292R Calypso–ASX proteins. Experiments were performed and recorded as described in **a**. Black arrows indicate the elution profiles of protein standards. **c** SEC-MALLS analysis of wild type and mutant Calypso–ASX complexes. Elution profiles of each protein complex at 50 μM monitored by differential refractive indices (lines); the weight averaged molecular masses (Mw) are shown with symbols. Dashed lines correspond to the size of the 1:1 (55 kDa, bottom) and 2:2 (110 kDa, top) Calypso–ASX complex (left). Cartoon representation of the Calypso–ASX complex (right), coloured as in Fig. 1b. A close-up view showing the packing of the two coiled-coil hairpins, with residues that mediate dimerisation indicated (see Supplementary Fig. 5a, b). **d** Overlay of experimental SAXS data (black circles) and scattering profile calculated using CRYSOL (ref. [59]) for the crystal structure of Calypso–ASX. Agreement between the experimental data and calculated scatter pattern is signified by Chi = 0.562 (see Supplementary Fig. 5a, f, g, and Supplementary Table 5). **e** Assays comparing the appearance of crosslinked species in wild type and L340A Calypso–ASX complexes at increasing concentrations of BS³. Samples were resolved by reducing SDS-PAGE gel and stained with Coomassie Blue. BS³, bis(sulfosuccinimidyl)suberate (see Supplementary Fig. 5h)

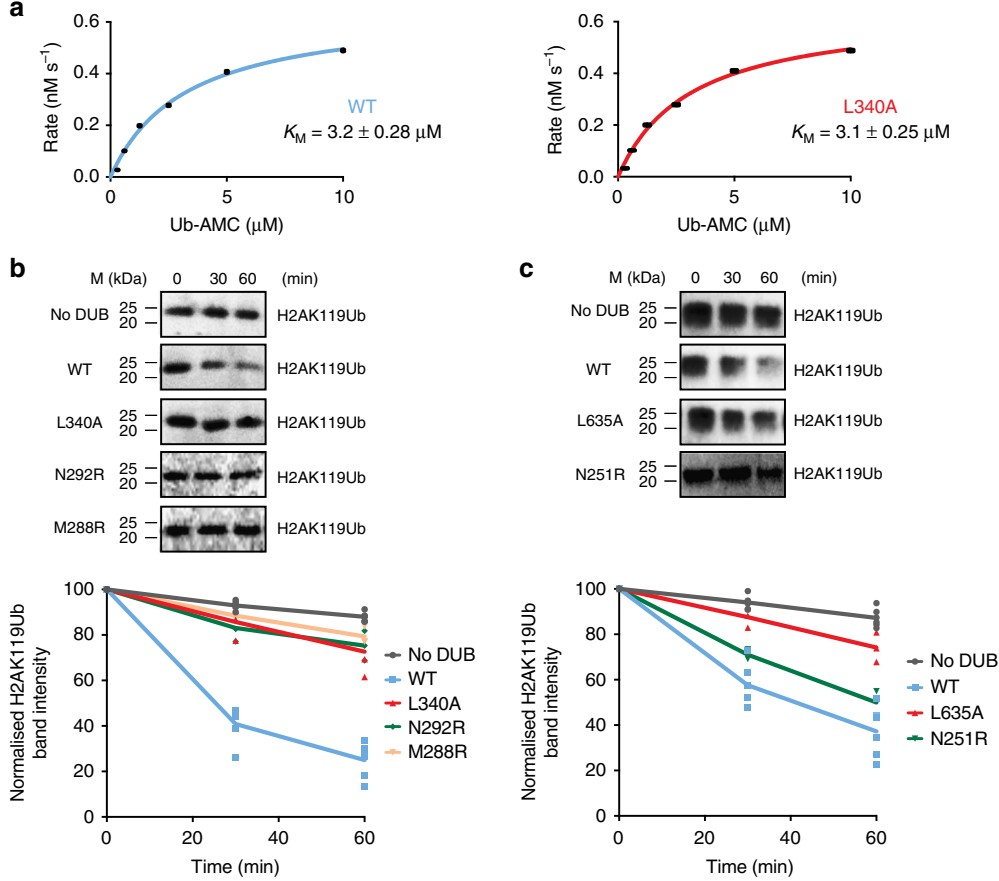

**Fig. 5** A higher-order PR-DUB oligomer promotes activity on nucleosomes. **a** Michaelis–Menten analysis of wild type and L340A Calypso–ASX proteins cleavage of Ubiquitin-AMC. Error bars indicate ±SEM ($n = 2$ independent experiments) (see Supplementary Fig. 3b). **b** Activity assays comparing the ability of wild type, L340A, N292R, and M288R Calypso–ASX complexes to remove ubiquitin from nucleosomes mono-ubiquitinated at H2AK119. 5-IAF-labelled ubiquitin was used to allow visualisation (top). Bands corresponding to H2AK119Ub nucleosomes were quantified. Quantitated sextuplets (for No DUB and WT reactions) or triplicates (for all Calypso–ASX mutant reactions) corresponding to six or three independent experiments are shown below, with a line passing through the mean. **c** Activity assays comparing the ability of wild type, L635A, and N251R BAP1–ASXL1 complexes to cleave ubiquitin from H2AK119Ub nucleosomes. Assays were visualised and quantified as described in **b**

To determine if the bidentate structure is a common feature of other PR-DUB complexes, we generated an alignment of the coiled-coil hairpin of nine different BAP1 and Calypso-like DUB sequences (Supplementary Fig. 6a). Consistent with an important role in oligomerisation (Fig. 4a–c), the residues corresponding to Calypso Leu340, Met288, and Asn292 are highly conserved throughout Calypso/BAP1-type DUBs. In order to determine if the coiled-coil hairpin is functionally conserved in the human PR-DUB complex, the residue in BAP1 corresponding to Calypso Leu340 (i.e. Leu635) was mutated to alanine. Wild type and mutant BAP1–ASXL1 complexes were first compared in Ubiquitin-AMC cleavage assays. As previously shown for the corresponding Calypso–ASX proteins, we did not detect a noticeable difference in $K_M$ between the two BAP1–ASXL1 constructs (Supplementary Fig. 6b), demonstrating that mutation at Leu635 did not induce structural changes in BAP1 that could affect its catalytic activity. Wild type and L635A BAP1–ASXL1 complexes were then tested for their ability to cleave ubiquitin from H2AK119Ub nucleosomes. Consistent with a conserved role for Calypso/BAP1 dimerisation, wild-type BAP1–ASXL1 was able to deubiquitinate H2AK119Ub, whereas the BAP1 (L635A)–ASXL1 complex had decreased activity (Fig. 5c). Mutation of the BAP1 residue corresponding to Calypso Asn292 (i.e. Asn251) showed a moderate reduction in the

catalytic activity of the BAP1(N251R)–ASXL1 complex compared to wild-type protein (Fig. 5c).

Overall, these analyses demonstrate that oligomerisation of Calypso–ASX via the coiled-coil region in Calypso is required for deubiquitination of H2AK119Ub, and that the same surface is required for efficient activity of BAP1–ASXL1. We next sought to explain why the bidentate 2:2 complexes acted more efficiently on H2AK119Ub nucleosomes even though the inherent enzymatic activities of 1:1 or 2:2 complexes are equivalent (Fig. 5a and Supplementary Fig. 6b).

**PR-DUB oligomerisation enables efficient nucleosome binding.** Sahtoe et al.[9] recently reported that the C-terminal positively charged tail of BAP1 is crucial for recruitment of the human PR-DUB complex to nucleosomes. Calypso also contains a homologous stretch of positively charged residues C-terminal to its ULD domain (Fig. 1a and Supplementary Fig. 1a, c). We hypothesised that an oligomeric DUB complex—containing two copies of either BAP1 or Calypso—would have two positively charged tails to promote recruitment of the PR-DUB to nucleosomes. Accordingly two suitably oriented tails would increase avidity for nucleosomes and thus explain increased activity relative to basal, monodentate, PR-DUB activity, as shown in Fig. 5.

To first verify if the C-terminal positively charged tail of Calypso is required for efficient recruitment of the *Drosophila* PR-DUB to nucleosomes, we performed electrophoretic mobility shift assays. While wild-type Calypso–ASX lacking the C-terminal tail (here referred to as WT$^{no\ tail}$) did not bind nucleosomes, the wild-type complex bearing the C-terminal tail (referred to as WT) was capable of shifting the nucleosome core particles in a concentration-dependent manner (Fig. 6a). In addition, we tested wild-type Calypso$^{no\ tail}$–ASX and BAP1$^{no\ tail}$–ASXL1 complexes for their ability to cleave ubiquitin from nucleosomes mono-

ubiquitinated at H2AK119. Consistent with our binding experiments, as well as with a previous study by Sahtoe et al.[9], both complexes showed impaired activity (Supplementary Fig. 6c). This indicates that, similarly to BAP1, the positively charged tail in Calypso is essential for interaction of the *Drosophila* PR-DUB with nucleosomes.

Having established the importance of the C-terminal tail in Calypso–ASX recruitment to nucleosomes, we then investigated if the Calypso-mediated bidentate PR-DUB complex was required for binding to its biological substrate. Consistent with activity

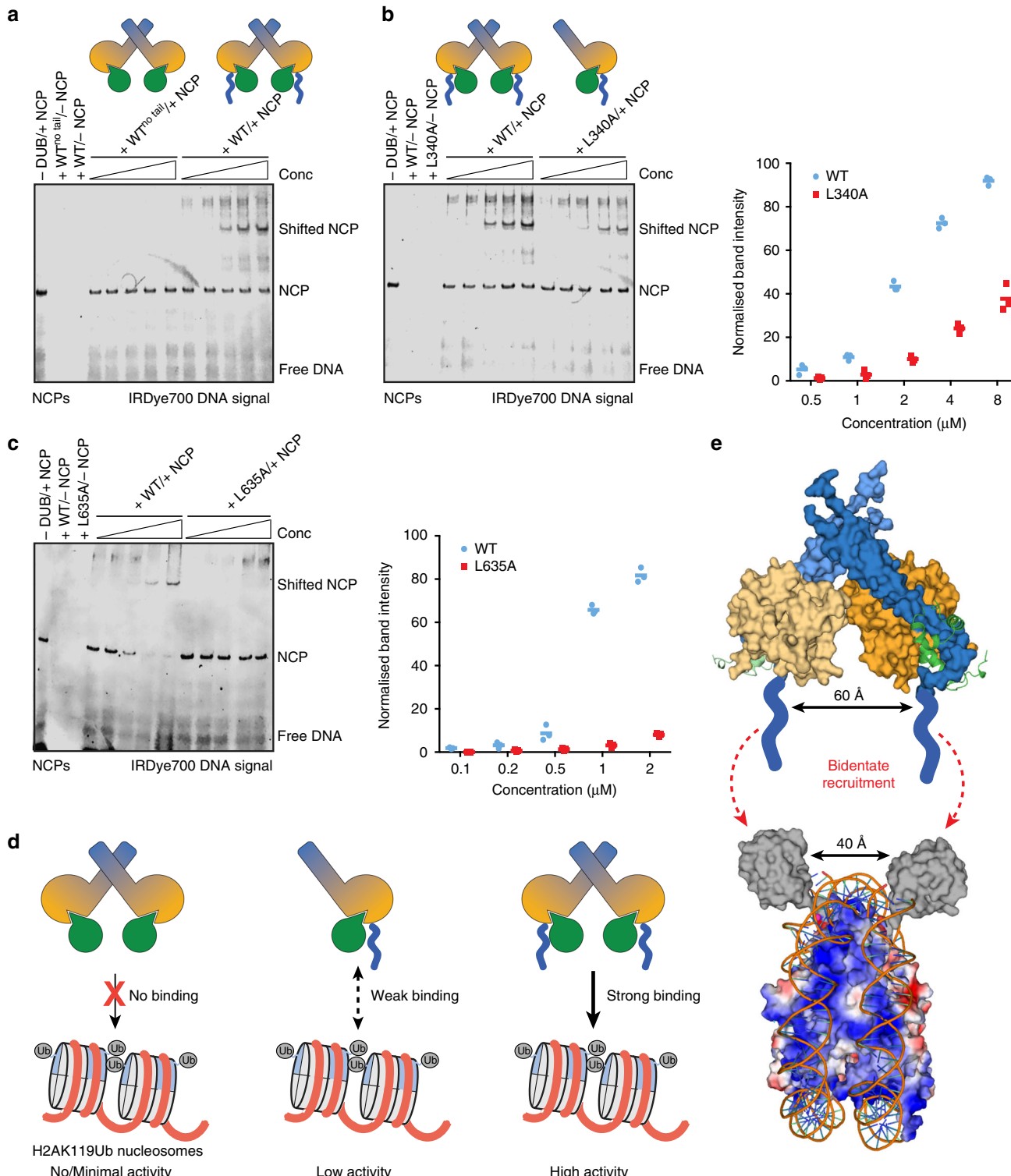

assays (Fig. 5b), wild-type Calypso–ASX was able to efficiently retard mobility of the nucleosome band, whereas the Calypso (L340A)–ASX complex had reduced ability to bind nucleosomes and therefore induced less nucleosome mobility shift at equivalent concentrations (Fig. 6b). Performing equivalent experiments with the corresponding BAP1–ASXL1 complexes demonstrated that wild-type BAP1–ASXL1 readily shifted the nucleosome band, whereas the L635A BAP1–ASXL1 complex had a dramatically impaired ability to induce nucleosome mobility shift at the same concentrations (Fig. 6c).

These experiments are consistent with the hypothesis that a bidentate 2:2 PR-DUB complex is most efficiently recruited to nucleosomes by virtue of having two positively charged C-terminal tails, leading to enhanced activity and efficient removal of H2AK119Ub (Fig. 6d). The bidentate Calypso–ASX complex from the crystal structure places two C-terminal tails on the same side of the PR-DUB, with two DUB active sites approximately 60 Å apart. Given the symmetry of histone proteins within a nucleosome—with two Histone 2A molecules sitting on opposite sides of the DNA but generally adjacent one another—a bidentate PR-DUB is well-suited for recruitment to individual nucleosomes, and subsequent removal of ubiquitin marks (Fig. 6e).

**Model of PR-DUB regulation by deubiquitinase oligomerisation.** The finding that a bidentate PR-DUB complex with two DUB components has enhanced activity on nucleosomes allows us to propose a revised model for regulation of PR-DUB activity (Fig. 7a). Because DUB–ASX interactions are relatively tight ($K_D$ ~6–18 nM)[16,17] and bidentate complex assembly is relatively low affinity (dissociating in the 1–18 µM range; Fig. 3c), it is likely that the PR-DUB predominantly exists as a 1:1 complex in the nucleus with low basal activity on H2AK119Ub nucleosomes. Enrichment of the 1:1 complex in specific regions of the genome, either by direct interaction of the PHD domain with nucleosomes or via indirect recruitment of the PR-DUB by other binding partners, would enhance the local concentration such that DUB (Calypso/BAP1) oligomerisation and bidentate higher-order complex formation becomes favoured. Once the bidentate complex forms, the two C-terminal tails of BAP1/Calypso are brought into an optimal orientation for enhanced nucleosome recruitment and removal of H2AK119Ub. Through such a mechanism maximal PR-DUB activity can be focussed specifically within the genome. Conversely, activity of the PR-DUB outside of these foci is low, and can be antagonised by PRC1-based H2A ubiquitination.

The proposed model accounts for distinct classes of human PR-DUB mutations that drive carcinogenesis. For instance, ASXL1 truncations that lead to a loss of the C-terminal PHD domain (Fig. 7b, [1]) can globally erase H2AK119Ub[39], potentially because these ASXL1 molecules can moderately activate BAP1 with their Deubad domains but fail to locally concentrate the PR-DUB in particular areas of the genome. Heterozygous loss of BAP1 (Fig. 7b, [2]) would reduce overall levels of BAP1 protein, disfavouring bidentate complex assembly and decreasing specific activity. In a similar vein, mutations that truncate the BAP1 C-terminal tail (Fig. 7b, [3]) would disfavour efficient nucleosome recruitment, even in the presence of two BAP1 protomers. Finally, missense mutations such as those depicted in Fig. 2 would directly block activity of either monodentate or bidentate complexes (Fig. 7b, [4]) and disrupt homoeostasis of genome-wide H2AK119Ub.

## Discussion

The PR-DUB complex plays a vital role in regulating H2AK119Ub histone marks, and mutations in human PR-DUB genes lead to a spectrum of malignancies. Here we present the first PR-DUB structure from any organism. The structure provides two major insights—it clarifies how the Deubad protein ASX (and its human homologues ASXL1–3) stimulates the deubiquitinase activity of Calypso (and BAP1 in humans); and provides a model to explain how the Calypso–ASX complex forms higher-order oligomers that are crucial for its biological activity. Both aspects contribute to a more complete understanding of how PR-DUB activity is regulated in multicellular organisms, and how mutations in the PR-DUB disrupt normal regulation to drive cancer development.

This work provides a structural basis upon which to interpret previous studies showing that the ASXL1/2 Deubad domains bind tightly to BAP1, and activate the complex by forming a composite binding site for ubiquitin[8,9]. The structure of Calypso–ASX is reminiscent of the activated deubiquitinase complex observed for UCH-L5 bound to the Deubad domain of Rpn13 (Supplementary Fig. 2, left panel) (PDB codes: 4UEM and 4WLQ), rather than the inhibitory Deubad INO80G bound to the same UCH DUB (Supplementary Fig. 2, right panel) (PDB codes: 4UF5 and 4WLP)[16,17]. Because UCH-L5–Rpn13 structures with and without ubiquitin bound are available, we could model an active intermediate with ubiquitin bound in the cleft of Calypso–ASX (Fig. 1c). Mutagenesis analyses performed in this study (Fig. 2c and Supplementary Fig. 3b, c), as well as in previous works[8,9,18]

**Fig. 6** PR-DUBs oligomerisation enables efficient nucleosome recruitment. **a** Electrophoretic mobility shift assays comparing the ability of wild-type Calypso[no tail]–ASX and Calypso–ASX complexes to bind nucleosomes at varying protein concentrations (0.5, 1, 2, 4, 8 µM). Recombinant nucleosomes reconstituted with a 220 bp Widom 601 DNA sequence labelled at the 5' with an IRDye®700 were used to visualise nucleosomes. **b** Electrophoretic mobility shift assays testing the binding of wild type and L340A Calypso–ASX proteins to nucleosomes at increased protein concentrations (as for panel **a**). Bands corresponding to the shifted nucleosome particle (indicated as shifted NCP) were quantified. Quantitated triplicates corresponding to three independent experiments are shown on the right, with a line passing through the mean. **c** Electrophoretic mobility shift assays comparing the ability of wild type and L635A BAP1–ASXL1 complexes to bind nucleosomes at different protein concentrations (0.1, 0.2, 0.5, 1, and 2 µM). Recombinant nucleosomes were reconstituted as outlined in **a**. Experiments were quantified as described in **b**. **d** Schematic representation of the mechanisms that underpin PR-DUB recruitment and activity on nucleosomes. In the absence of the C-terminal positively charged tails (left), the oligomeric PR-DUB complex cannot be recruited to nucleosomes, resulting in a total loss of activity and maintenance of the H2AK119Ub mark. The 1:1 PR-DUB complex (middle), despite the presence of the C-terminal tail, has reduced affinity for nucleosomes and hence limited ability to deubiquitinate H2AK119Ub. Only a PR-DUB complex that bears two C-terminal tails and that has full ability to form a bidentate complex (right) can be efficiently recruited to nucleosomes, thereby resulting in increased activity. **e** Scale model for bidentate recruitment of the Calypso–ASX complex to H2AK119Ub nucleosomes. The Calypso dimer and ubiquitin are shown in surface representation, while the ASX Deubad domains are shown as cartoon; the surface of the nucleosome particle is represented as electrostatic potential. The C-terminal positively charged tails of the two Calypso molecules are shown in blue. Black arrows indicate the distances between the two Calypso active sites as well as between the two mono-ubiquitinated Histone 2A Lys119 residues of the nucleosome particle

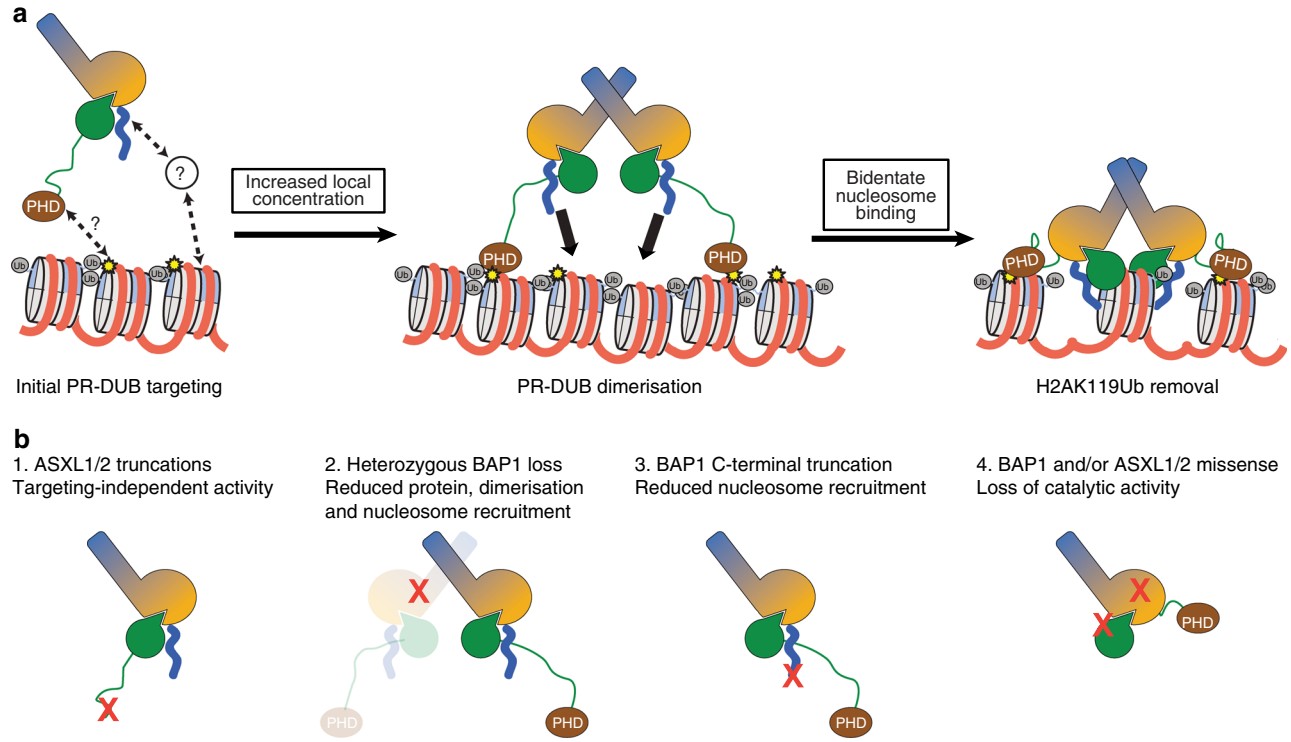

**Fig. 7** Model for PR-DUB regulation and its disruption in human cancers. **a** The 1:1 PR-DUB complex is first targeted to nucleosomal arrays either via the PHD domain of ASX-like proteins or through interactions with transcription factors and/or other binding partners (left). Enrichment of multiple 1:1 complexes in specific genomic regions increases the local PR-DUB concentration, favouring Calypso/BAP1 dimerisation and formation of a bidentate PR-DUB complex (middle). With bidentate complex formation, two C-terminal positively charged tails of Calypso/BAP1 are optimally oriented, enhancing nucleosome recruitment and efficient removal of H2AK119Ub (right). **b** Distinct classes of human PR-DUB mutations can impair PR-DUB targeting, recruitment, and activity on nucleosomes via multiple mechanisms

demonstrate that mutations at the composite interface between DUB, Deubad, and ubiquitin have a dramatic effect on the ability of the PR-DUB to interact with ubiquitin, ultimately leading to impaired activity. These findings, coupled with the recurrence of the same mutations in human cancers[25,26] (Supplementary Table 3), underline the high sensitivity of this juncture for PR-DUB activity.

A striking aspect of this study is that Calypso–ASX forms stable 2:2 oligomers in solution, in which two Calypso–ASX units interact via the coiled-coil region of Calypso. Several lines of evidence suggest that the human PR-DUB also forms higher-order oligomers, and a conserved mode of deubiquitinase oligomerisation is consistent with our finding that mutating the coiled-coil of BAP1 reduces activity on H2AK119Ub nucleosomes (Fig. 5b, c). In vitro experiments have previously shown that BAP1 oligomerises and BAP1–ASXL1 can form a complex with apparent 2:1 stoichiometry[9,37], while mass spectrometry and biochemical studies have shown that the PR-DUB can consist of BAP1–ASXL1 or BAP1–ASXL2 complexes[8,37]. It remains to be seen whether the equimolar (2:2) DUB–ASX stoichiometry we observe is conserved in humans but obscured by redundancy of human PR-DUB components. There is also scope for the BAP1-specific insert to influence the makeup of the PR-DUB in humans, or its stoichiometry, relative to the simpler two-component system of *Drosophila*. Our structural and biochemical analyses now show that a major effect of PR-DUB dimerisation is to place two positively charged tails in close proximity, mediating bidentate nucleosome recruitment. Such an arrangement is crucial to promote higher affinity interaction with nucleosomes by increasing avidity, and ultimately allowing

efficient removal of H2AK119Ub (Figs. 6d and 7a). Intriguingly, the arrangement of the Calypso dimer places the two active sites approximately 60 Å apart, which would allow it to span a single mononucleosome and access two Histone 2A Lys119 residues (Fig. 6e). While spanning a single nucleosome is attractive, it is also possible that the bidentate complex bridges multiple nucleosomes. In either scenario increased local concentration of the PR-DUB would drive oligomerisation and efficient removal of H2AK119Ub marks from specific genomic foci, thereby leading to decompaction of the chromatin fibres and enhancement of gene expression.

BAP1 and ASXL1/2 undergo both missense and nonsense mutations in various cancers and exhibit hallmarks of haploinsufficiency, whereby mutations even in a heterozygous form have deleterious effects[20]. In this sense, the requirement of higher-order PR-DUB oligomerisation for efficient activity on nucleosomes is highly relevant to cancer epigenetics. The moderate affinity of Calypso dimerisation suggests that activity could be particularly sensitive to protein concentration—mutation of one gene copy could effectively deplete the activity of the remaining translated wild-type protein. A major outstanding question within the model illustrated in Fig. 7a is the exact mechanism that drives PR-DUB enrichment in particular regions of the genome. Several factors could be at play: for instance (i) the C-terminal PHD domain of ASX-like proteins may recognise specific histone marks; or (ii) the PR-DUB may be localised indirectly by other binding partners. The former mechanism is suggested by the general role of PHD domains as histone readers and studies demonstrating dysregulation of H2AK119Ub upon truncation of the ASXL1/2 PHD domains;[21,22,40] the latter

mechanism is supported by data showing significant enrichment of BAP1 at E-twenty six (Ets) and Specific protein 1 (Sp1) transcription factor-binding sequences[32] or that ASXL1 participates in direct recruitment of the PRC2 complex to specific genomic loci[17]. We anticipate future structural and biochemical studies will enable further insight. Finally, it remains to be seen how the bidentate complex may affect deubiquitination of other PR-DUB substrates—for instance BAP1 has been proposed to remove ubiquitin from its own C-terminal tail, as well as Host Cell Factor-1 (HCF-1; which binds within the BAP1-specific insert) and Ube2O[29,30,34], both of which are highly relevant to the role of the PR-DUB in cancers.

In summary, this study provides a template for understanding PR-DUB function in both normal and cancer biology. We show that the *Drosophila* PR-DUB complex provides an excellent structural model to understand the impact of missense mutations in the human complex, and that a 2:2 PR-DUB complex is markedly more efficient at removing H2AK119Ub—the latter finding suggests a mechanism by which PR-DUB is focussed in particular genomic regions in healthy cells, and disrupted by multiple mechanisms to drive malignancy.

## Methods

**Plasmids and cloning**. To generate a co-expression construct for Calypso–ASX, the DNA sequences of Calypso (corresponding to residues 43–404) and ASX (residues 207–340) were fused together with the linker sequence that connects the first and the second multiple cloning site in the pET-Duet1 vector (Novagen). Residues 258–270 in ASX, which are not conserved in the human ASXL proteins and are predicted to be disordered, were omitted from this co-expression construct. 5′-(CAGGGACCCGGT) and 3′-(TAACCGGGCTTCTCCTCG) overhangs, required for Ligation-Independent Cloning (LIC), were also introduced. The resulting co-expression construct was synthesised by Integrated DNA Technology (IDT) and cloned into an N-terminal His$_6$ tag expression vector of the NKI LIC Suite[41]. The same construct was subsequently used to introduce the C-terminal positively charged tail in Calypso. Briefly, site-directed mutagenesis was first employed to replace the STOP codon of Calypso with a *Kpn*I restriction site. Following linearisaton of the vector, the DNA sequence corresponding to the positively charged tail of Calypso (residues 405–471; synthesised by IDT) was introduced by Gibson assembly[42]. A co-expression construct encoding the RING domains of mouse Ring1B (residues 1–159) and Bmi1 (residues 1–109)[39] was synthesised and cloned as described for Calypso–ASX, with the exception that an N-terminal glutathione-*S*-transferase (GST)-His$_6$ tag expression vector was used. Ubiquitin-H2A$^{tail}$, in which the sequence of wild-type Ubiquitin was linked to the C-terminal tail of Histone 2A (residues 120–130), was synthesised by IDT and cloned into pET-3a (Novagen) using *Nde*I and *Bam*HI. Constructs encoding the human E1 enzyme Ube1 (Addgene plasmid # 34965)[43] and *Xenopus laevis* histones (Addgene plasmid # 66890)[44] were purchased from Addgene, while the Widom 601 DNA sequence[45] was synthesised by IDT. The sequence of the Widom 601 DNA used in this study is as follows (randomly chosen nucleotide sequences are in small case letters while the *Eco*RV restriction sites are underlined and in italics):

gcgtaa*GATATC*ATCGATGGACCCTATACGCGGCCGCCCTGGAGAATCC CGGTGCCGAGGCCGCTCAATTGGTCGTAGACAGCTCTAGCACCGCTTAA ACGCACGTACGCGCTGTCCCCCGCGTTTTAACCGCCAAGGGGATTACT CCCTAGTCTCCAGGCACGTGTCAGATATATACATCCTGTGCATGTATTGA ACAGCGACCTGAT*GATATC*gcgtaa

Wild-type Ubiquitin was cloned into pET-3a using *Nde*I and *Bam*HI[46]. To allow labelling of Ubiquitin with 5-iodoacetamidofluorescein (5-IAF) (Thermo Fisher Scientific Molecular Probes), site-directed mutagenesis was used to introduce a methionine and cysteine residues before the beginning of the Ubiquitin sequence, resulting in Met-Cys-Ubiquitin[47]. UbcH5b$^{C21S/C107S/C111S}$ (3C/S) was cloned into pGEX-6P-3 (GE Healthcare) with *Bam*HI and *Eco*RI restriction sites.

To generate constructs for insect cells expression, the DNA sequences of Calypso (residues 43–471) and BAP1 (residues 1–729 or 1–712) were cloned into a pFastBac expression vector modified to incorporate an N-terminal His$_6$ tag followed by a 3C protease cleavage site with LIC compatible overhangs, while the DNA sequences corresponding to the Deubad domains of ASX (residues 207–340) and ASXL1 (residues 247–366) were cloned into an equivalent N-terminal StrepII tag pFastBac vector. Calypso and BAP1 dimer interface mutants (i.e. Calypso L340A, M288R and N292R; BAP1 L635A and N251R) were cloned in the same manner as wild-type constructs. Positive clones were selected, and the corresponding wild type and mutant Calypso–ASX and BAP1–ASXL1 complexes assembled into pBig1a for co-expression by Gibson assembly[42]. Following transformation of the Gibson mixtures into DH10BAC cells (Life Technologies), a second round of selection was used to isolate the clones containing the assembled complexes.

All point mutations were introduced using the one-step site-directed mutagenesis protocol[48].

**Protein expression and purification**. Calypso–ASX wild type and mutant complexes were co-expressed with an N-terminal His$_6$ tag in *E. coli* BL21 (DE3) cells (Novagen). Cells were grown at 37 °C in Luria-Bertani (LB) medium to an OD$_{600}$ of approximately 0.6, then induced with 0.2 mM isopropyl-β-D-1-thiogalactopyranoside (IPTG) and grown at 18 °C overnight (O/N). Bacterial pellets were resuspended in 50 mM HEPES pH 7.5, 10% (v/v) glycerol, 10% (w/v) sucrose, 50 mM NaCl, and 5 mM imidazole. Cells were lysed by sonication using a Sonifier (Heat Systems Ultrasonics) instrument and the soluble fractions bound to Ni$^{2+}$-nitrilotriacetic acid (NTA) resin (Sigma-Aldrich) for 1 h at 4 °C on a rotary mixer. After binding, the resins were washed four times with 50 mM HEPES pH 7.5, 10% (v/v) glycerol, 10% (w/v) sucrose, 300 mM NaCl, and 5 mM imidazole. Protein complexes were eluted with the same buffer containing 300 mM imidazole. To release Calypso variants from the His$_6$ tag, the recovered fractions were combined and 3C protease and 5 mM dithiothreitol (DTT) added. Samples were incubated O/N at 4 °C and cleavage was monitored by SDS-PAGE analysis. Calypso–ASX complexes were then diluted ten times in 50 mM HEPES pH 7.5 and 5 mM NaCl, and loaded onto a 5 mL HiTrap QHP column (GE Healthcare) pre-equilibrated in the same buffer. A linear 25 mL gradient, from 0.005 to 1 M NaCl, was used to elute the proteins. The recovered Calypso–ASX complexes were finally purified on a Superdex 200 10/300 column (GE Healthcare) in 20 mM HEPES pH 7.5 and 100 mM NaCl.

For the expression and purification of the protein complexes derived from insect cells, the Bac-to-Bac baculovirus expression system (Invitrogen) was used to generate baculovirus stocks for wild type and mutant Calypso–ASX and BAP1–ASXL1 proteins according to the manufacturer's protocol, with the exception that FuGENE 6 (Promega) was used for transfection. The *Trichoplusia ni* (*Tni*) (Expression Systems) cell line was infected with the obtained baculovirus stocks co-expressing wild type and dimer interface mutants (i.e. L340A, M288R, and N292R) Calypso$^{43–471}$–ASX$^{207–340}$ complexes, as well as with baculoviruses for the co-expression of the N251R BAP1$^{1–729}$–ASXL1$^{247–366}$ and wild-type BAP1$^{1–712}$–ASXL1$^{247–366}$ (later referred to as BAP1$^{no\ tail}$–ASXL1) proteins. *Spodoptera frugiperda* (*Sf*) 9 (Expression Systems) cells were used for the co-expression of the wild type and L635A complexes in BAP1$^{1–729}$–ASXL1$^{247–366}$. Cells were infected in log phase growth at a density of approximately $1.0 \times 10^6$ cells/mL. Following infection, cells were grown in ESF921 medium (Expression Systems) and incubated at 27 °C, 125 r.p.m., for either 48 h (*Tni*) or 72 h (*Sf*9) before being harvested. Cell pellets were resuspended in 50 mM Tris-HCl pH 8.0, 10% (v/v) glycerol, 150 mM NaCl, and 10 mM imidazole supplemented with 10 µg DNaseI (Applichem). Following incubation on ice for 30 min, cells were lysed using a Sonics (Vibra Cell) instrument, and the soluble fractions bound to Ni$^{2+}$-NTA resin as described above. Wild type and mutant Calypso–ASX and BAP1–ASXL1 complexes were eluted with 50 mM Tris-HCl pH 8.0, 10% (v/v) glycerol, 150 mM NaCl, and 500 mM imidazole. Recovered fractions were pooled and dialysed O/N against 20 mM HEPES pH 7.5 and 100 mM NaCl.

The Ring1B–Bmi1 complex was co-expressed with an N-terminal GST-His$_6$ tag in *E. coli* Rosetta 2 (DE3) cells (Novagen). Cells were grown at 37 °C in LB medium until OD$_{600}$ reached 0.6, then induced with 0.2 mM IPTG and 0.1 mM ZnCl$_2$. After induction, cells were grown at 28 °C for 4 h. Bacterial pellets were resuspended in 50 mM MES pH 6.5 and 200 mM NaCl, lysed, and the soluble fractions bound to glutathione (GSH) Sepharose resin (GE Healthcare) for 1 h at 4 °C on a rotary mixer. The GST-His$_6$ tag was removed by O/N incubation with 3C protease and 5 mM DTT. Upon completion of the cleavage, the Ring1B–Bmi1 complex was washed off the resin using a hand-held column (Bio-Rad) and purified on a Superdex 75 10/300 column (GE Healthcare) in 20 mM MES pH 6.5 and 100 mM NaCl.

Ubiquitin-H2A$^{tail}$ was expressed as untagged construct in *E. coli* BL21 (DE3) cells. Cells were grown at 37 °C in LB medium to an OD$_{600}$ of approximately 0.6, then induced with 0.2 mM IPTG and grown at 37 °C for 4 h. Bacterial cells were sonicated in 30 mM MES pH 6.5, 10 mM NaCl, and 1 mM ethylenediaminetetraacetic acid (EDTA), and the soluble protein fractions loaded onto a 5 mL HiTrap SPHP column (GE Healthcare) pre-equilibrated in the same buffer used for lysis. The fusion protein was then eluted with a linear 25 mL gradient, from 0 to 1 M NaCl, and further purified on a Superdex 75 10/300 column (GE Healthcare) in 20 mM Tris-HCl pH 7.5 and 100 mM NaCl.

Human Ube1 was expressed with an N-terminal His$_6$ tag in *E. coli* BL21 (DE3) cells[43]. Cells were grown at 37 °C in LB medium to an OD$_{600}$ of 0.6, then induced with 0.2 mM IPTG, and grown at 18 °C O/N. Bacterial pellets were lysed in 20 mM Tris-HCl pH 8.5, 100 mM NaCl and 5 mM imidazole, and the supernatant was loaded onto a 5 mL HisTrap column (GE Healthcare). A linear 50 mL gradient, from 5 to 250 mM imidazole, was then applied. Fractions containing hUbe1 were recovered and finally purified on a HiLoad Superdex 200 16/60 column (GE Healthcare) in 20 mM Tris-HCl pH 7.5 and 150 mM NaCl.

UbcH5b$^{3C/S}$ was expressed with an N-terminal GST tag in *E. coli* BL21 (DE3) cells. Cells were grown at 37 °C in LB medium until OD$_{600}$ reached 0.6, then induced with 0.2 mM IPTG and grown at 37 °C for 4 h. Bacterial pellets were lysed in phosphate-buffered saline (PBS; 10 mM Na$_2$HPO$_4$, 1.8 mM KH$_2$PO$_4$, 140 mM NaCl, and 2.7 mM KCl, pH 7.4), and the soluble fractions bound to GSH Sepharose resin for 1 h at 4 °C on a rotary mixer. The GST tag was removed by O/N

incubation with 3C protease and 5 mM DTT. Upon completion of the cleavage, UbcH5b[3C/S] was recovered and further purified on a HiLoad Superdex 75 16/60 column (GE Healthcare) in PBS.

Wild-type and Met-Cys-Ubiquitin variants were expressed as untagged constructs in *E. coli* BL21 (DE3) cells[46]. Cells were grown at 37 °C in LB medium to an $OD_{600}$ of 0.6, then induced with 0.2 mM IPTG and grown at 37 °C for 4 h. Bacterial cells were sonicated in 50 mM $NH_4C_2H_3O_2$ pH 4.5 and 1 mM EDTA, and the supernatant was incubated for 1 h at 60 °C. Following clarification, the soluble fraction was purified on a 5 mL HiTrap SPHP column and proteins were eluted with a linear 25 mL gradient, from 0 to 1 M NaCl. Wild-type Ubiquitin used in pulldown experiments was purified on a HiLoad Superdex 75 16/60 column in 20 mM Tris-HCl pH 7.5 and 150 mM NaCl. Whereas Met-Cys-Ubiquitin used in subsequent labelling steps was purified on the same column in HBS buffer (50 mM HEPES pH 7.4, 150 mM NaCl, and 5 mM $MgCl_2$). To allow labelling[47] with 5-IAF, the peak fractions were pooled and incubated with 4 mM 5-IAF for 1 h at room temperature. The labelled protein was then separated from excess dye by using a 10 mL HiTrap desalting column (GE Healthcare), pre-equilibrated in HBS buffer. Labelled Ubiquitin was incubated O/N at room temperature to allow unconjugated protein to form disulphide-linked dimers. The mixture was finally separated on a HiLoad Superdex 75 16/60 column in HBS.

**Purification of recombinant histones.** Expression and purification of recombinant histones was essentially carried out as described by Shim et al.[44] Briefly, all four recombinant *Xenopus laevis* histones were co-expressed with N- and C-terminal $His_6$ tags in *E. coli* BL21 (DE3) pLysS cells (Novagen). Cells were grown at 37 °C in 2xYeast Extract Tryptone (YT) medium until $OD_{600}$ reached 0.4, then induced with 0.4 mM IPTG and grown at 37 °C for 20 h. Bacterial cells were resuspended in 20 mM Tris-HCl pH 8.0, 2 M NaCl, 1 mM phenylmethylsulfonyl fluoride (PMSF) and 0.5 mM tris(2-carboxyethyl)phosphine (TCEP), lysed, and the soluble fractions loaded onto a 5 mL HisTrap column (GE Healthcare) pre-equilibrated in the same buffer without PMSF. Following an initial washing step with 20 mM Tris-HCl pH 8.0, 2 M NaCl, 30 mM imidazole, and 0.5 mM TCEP, the octameric histone complex was eluted with the same buffer plus 500 mM imidazole. Fractions containing equivalent amounts of all four histone proteins were pooled, and a corresponding volume of 20 mM Tris-HCl pH 8.0 and 0.5 mM TCEP added in order to obtain a final salt concentration of 1 M. Thrombin protease (GE Healthcare) at a concentration of 1 U per mg of protein was added to the histone sample, and cleavage was carried out at room temperature for 48 h. Upon completion of the digestion, the octameric histone complex was purified on a Superdex 200 10/300 Increase column (GE Healthcare) and stored in 20 mM Tris-HCl pH 8.0, 2 M NaCl and 0.5 mM TCEP.

**Nucleosome reconstitution.** For the reconstitution of recombinant nucleosomes used in activity assays, a 220 bp Widom 601 DNA sequence was first amplified by polymerase chain reaction (PCR) using unlabelled forward (gcgtaa*GA-TATC*ATCGATGGACCCTAT) and reverse (ttacgc*GA-TATC*ATCAGGTCGCTGTTC) primers (purchased from IDT). For the synthesis of nucleosomes used in electrophoretic mobility shift assays, the same primers described above were used with the exception that the forward primer was synthesised with an IRDye®700 fluorophore attached at the 5′-end of the sequence (purchased from IDT). Following purification of both PCR products by ethanol precipitation and determination of the DNA concentration, nucleosome reconstitution was carried out as described in the EpiMark™ Nucleosome Assembly Kit (New England Biolabs; https://www.neb.com/protocols/2012/06/04/epimark-nucleosome-assembly-kit-e5350). Briefly, 8 μM of histone octamer was mixed with 25 pmol of either unlabelled or labelled Widom 601 DNA in a total volume of 100 μL. The mixture was allowed to equilibrate for 1 h at room temperature, and subsequently dialysed against Tris-EDTA (TE) buffers (20 mM Tris-HCl pH 8.0, 1 mM EDTA and 1 mM DTT) containing 1, 0.6, and 0.25 M NaCl. Each dialysis step was carried out for 2–3 h at 4 °C, with the exception of the 0.6 M TE buffer dialysis which was performed O/N at 4 °C. Reconstituted nucleosomes were further dialysed against 20 mM Tris-HCl pH 8.0, 0.25 M NaCl and 1 mM DTT for 2–3 h at 4 °C, then subjected to a heat shift by incubation at 37 °C for 30 min[9] and finally stored at 4 °C.

**Crystallisation and structure determination.** Crystals of the Calypso[43–404]–ASX[207–340] complex (~4 mg/mL protein in 20 mM HEPES pH 7.5 and 100 mM NaCl) were grown in 60-well microbatch plates (Nunc InterMed, Denmark) at 18 °C. Purified Calypso[43–404]–ASX[207–340] was mixed with chymotrypsin at a final concentration of 1 μg/mL immediately prior to crystallisation. Plates were filled with 6 mL of Al's Oil (Hampton Research), and drops were set up manually by combining 1 μL of the purified complex with 1 μL of crystallisation solution containing 0.1 M Bis-Tris propane pH 8.5, 0.0125 M potassium iodide, 13% (w/v) PEG 3350 (Hampton Research) and 3% (v/v) MPD (Hampton Research). Crystals grown under this condition were used to prepare a seeding solution, which was then serially diluted (i.e. 1:10. 1:100, 1:1000, 1:10,000 and 1:100,000) into a new screen. Thicker crystals were obtained when the seeding stock was diluted 1:10 in 0.1 M Bis-Tris propane pH 8.5, 0.05 M potassium iodide, 13% (w/v) PEG 3350, and 10% (v/v) MPD. In this screen, drops were set up by manually mixing 2 μL of the purified complex with 2 μL of buffer solution. The

resulting Calypso[43–404]–ASX[207–340] crystals were harvested after one week and cryoprotected with 20% (v/v) glycerol before flash cooling in liquid nitrogen.

X-ray diffraction data were collected at 0.9537 Å, 100 K, on an Eiger 16M detector at the MX2 beamline of the Australian Synchrotron. Data were processed and scaled using XDS[49] and exhibited strong anisotropy; therefore, an anisotropy-corrected reflection file generated using the UCLA diffraction anisotropy correction server[50] was used for structure refinement. The structure was solved by molecular replacement with the programme Phaser-MR[51] using the coordinates of the UCH-L5–RPN13 complex (PDB code: 4UEM)[16] as a search model. Structure refinement was performed with PHENIX refine[52] using the TLS function available within the interface, while Coot[53] was used to iteratively build missing residues. The structure was solved in space group $I12_1$ at 3.5 Å resolution, with final $R_{work}$ and $R_{free}$ values of 24.4% and 29.4%, respectively. Ramachandran statistics show that 91.95% of residues are in favoured regions, 7.90% in allowed regions, and 0.14% are outliers. Statistics for data collection and refinement of the structure are summarised in Supplementary Table 1, while figures showing improvement of the electron density map following anisotropy correction are included in Supplementary Figure 5b–e.

**SAXS analysis.** SAXS data collection was performed at the Australian Synchrotron SAXS/WAXS beamline using an inline gel filtration chromatography setup[54,55]. Briefly, 60 μL of the purified Calypso[43–404]–ASX[207–340] complex at 8 mg/mL (145 μM) was injected onto an inline Superdex 200 5/150 column (GE Healthcare) and eluted at a flow rate of 0.2 mL/min via a 1.5 mm glass capillary positioned in the X-ray beam in 20 mM HEPES pH 7.5, 500 mM NaCl, 5% (v/v) glycerol, and 0.2 mM TCEP at 12 °C. Coflow SAXS was used to minimise sample dilution and maximise signal to noise[56].

Scattering data were collected in 2 s exposures over the course of the elution and 2D intensity plots with consistent scatter intensities from the peak of the SEC run were radially averaged, normalised to sample transmission, and background subtraction was performed using the Scatterbrain software (Stephen Mudie, Australian Synchrotron). Background scatter was assessed by averaging scattering profiles from earlier in the SEC run, before the protein was eluted. Guinier analysis of each scatter pattern across the single elution peak showed consistent radius of gyration ($R_g$) values, and superimposable patterns were averaged. Five profiles from the apex of the elution peak were averaged, and the background subtracted using Scatterbrain, to generate the averaged scatter patterns presented in the manuscript. Guinier data analyses were performed using PRIMUS[57]. Indirect Fourier transformation with GNOM[58] was used to obtain the distance distribution function, $P(r)$, and the maximum dimension, $D_{max}$, of the scattering particle. CRYSOL[59] was used to calculate theoretical scattering curves from the crystal structure atomic coordinates and compare them with experimental scattering curves. Statistics for data collection and analysis are reported in Supplementary Table 5.

**AUC analysis.** AUC experiments were conducted in a Beckman model XL-I instrument at a temperature of 20 °C. The purified Calypso[43–404]–ASX[207–340] complex was analysed at final concentrations ranging from 0.05 mg/mL (1 μM) to 2.2 mg/mL (40 μM) in 20 mM HEPES pH 7.8, 100 mM NaCl, and 0.8 mM TCEP.

For the sedimentation velocity experiment, 380 μL of sample and 400 μL of buffer solution were loaded into separate sectors of the same quartz cell and mounted in a Beckman 8-hole An-50 Ti rotor. Samples were centrifuged at a rotor speed of 50,000 r.p.m. Data were collected at a single wavelength in continuous mode, using a step-size of 0.003 cm without averaging. For the sedimentation equilibrium experiment, 100 μL of sample and 120 μL of buffer solution were loaded into separate sectors of the same quartz cell and mounted in a Beckman 8-hole An-50 Ti rotor. Samples were centrifuged at rotor speeds of 8000, 12,000, and 16,000 r.p.m. Data were collected at a single wavelength in step mode, using a step-size of 0.001 cm, averaging 20 data points.

Sedimentation velocity data at multiple time points were fitted to a continuous sedimentation-coefficient model[60,61] using the programme SEDFIT[62]. A van Holde–Weischet analysis[63] was performed for all runs, using the programme UltraScan III[64]. Sedimentation equilibrium data were fitted to a single species model with mass conservation, as implemented in SEDPHAT[65]. The error in the solution mass was estimated using the Monte-Carlo method as implemented in SEDPHAT. Solvent density (1.006 g/ml at 20 °C) and viscosity (0.01002 p), and an estimate of the partial specific volume of the protein (0.732 ml/g) were computed using the amino acid composition and the programme SEDNTERP[66].

**SEC-MALLS analysis.** To determine the oligomeric state of wild type and mutant Calypso–ASX variants, SEC coupled to multi-angle laser light scattering (SEC-MALLS) was used. Purified Calypso[43–404]–ASX[207–340] wild type and mutant proteins (in Fig. 3a, 60 μL of wild-type complex was used at 90 and 45 μM; in Fig. 4c, 60 μL of each protein complex was injected at a concentration of ~50 μM) were loaded at a flow rate of 0.15 mL/min onto a Superdex 200 5/150 GL Increase column (GE Healthcare) previously equilibrated in 20 mM HEPES pH 7.5 and 100 mM NaCl. The column was connected in line to a Dawn 8+MALLS detector (Wyatt Technology) and a Waters 410 differential refractometer (Millipore). Data were analysed using ASTRA version 5.3.4 software (Wyatt Technology).

**Analytical size-SEC analysis**. The elution profile of Calypso$^{43-404}$–ASX$^{207-340}$ wild type and mutant complexes was evaluated by injecting 60 µL of each purified protein (at ~30–50 µM) over a Superdex 200 5/150 GL Increase column (GE Healthcare) equilibrated in 20 mM HEPES pH 7.5 and 100 mM NaCl at a flow rate of 0.15 mL/min.

**Crosslinking experiments**. The functionality of the dimer interface in the Calypso–ASX complex was confirmed by using the amine cross-linker bis(sulfo-succinimidyl)suberate, BS$^3$ (Thermo Fisher Scientific). For time course experiments, purified Calypso$^{43-404}$–ASX$^{207-340}$ wild type and L340A complexes (10 µM each) were incubated with 0.5 mM BS$^3$ at room temperature for up to 60 min. For experiments with increased concentration of cross-linker, the same complexes were incubated with 0.25, 0.5, and 1 mM BS$^3$ at room temperature for 5 min. Reactions were terminated by adding 50 mM Tris-HCl pH 8.0 into each sample, followed by a second incubation step at room temperature for 15 min. All samples were resolved on 16% SDS-PAGE gels and visualised using Coomassie Blue staining. Uncropped gels for these experiments have been included in Supplementary Figs. 7 and 11.

**Ubiquitin-H2A$^{tail}$ assays**. Purified Ubiquitin-H2A$^{tail}$ (40 µM) was mixed with Calypso$^{43-471}$–ASX$^{207-340}$ wild type or the indicated mutant complexes (50 nM each) in assay buffer containing 25 mM HEPES pH 7.5, 150 mM NaCl, and 5 mM DTT. Reactions were incubated for 4 h at 37 °C in a final volume of 40 µL, and samples were taken at the indicated time points. Reactions were terminated by the addition of SDS-PAGE loading buffer containing β-mercaptoethanol. Proteins were separated on 14–20% SDS-PAGE gradient gels, and visualised using Coomassie Blue staining. Uncropped gels are provided in Supplementary Fig. 11.

**Ubiquitin-AMC assays**. The activity of Calypso$^{43-471}$–ASX$^{207-340}$ and BAP1$^{1-729}$–ASXL1$^{247-366}$ was determined by monitoring the release of fluorescent 7-amido-4-methylcoumarin (AMC) from the quenched Ubiquitin-AMC substrate, providing a direct readout of DUB activity. Purified Ubiquitin-AMC powder (UbiQ) was dissolved in pure DMSO to reach a concentration of 20 mg/mL, and then slowly diluted with milliQ water to a final concentration of 0.5 mg/mL. The residual amount of DMSO left in the reaction was never above 2.5%. When a single concentration of substrate was used, 0.5 nM of Calypso$^{43-471}$–ASX$^{207-340}$ wild type or the indicated mutant complexes was mixed with 1 µM of Ubiquitin-AMC in a final volume of 10 µL. Reactions were performed in assay buffer (25 mM HEPES pH 7.5, 150 mM NaCl, 5 mM DTT, and 0.05% Tween-20) at 25 °C using a black 384-well non-binding surface low flange Cliniplate (Labsystems). Measurements were taken every 20 s for 30 min in a CLARIOStar (BMG LABTECH) plate reader using 380 and 460 nm excitation and emission wavelengths, respectively. To determine kinetic parameters, 0.5 nM of the indicated Calypso$^{43-471}$–ASX$^{207-340}$ or BAP1$^{1-729}$–ASXL1$^{247-366}$ complexes were allowed to react with different concentrations of Ubiquitin-AMC (i.e. 0, 0.3, 0.6, 1.25, 2.5, 5, 10 µM) in 10 µL reactions. Assays were performed as described above. Fluorescence intensity units were converted to concentration of AMC released using free AMC (Sigma-Aldrich) as a standard. Initial rates were calculated as the slope of the linear part of the reaction curve, plotted against substrate concentration and fitted to the Michaelis–Menten equation using non-linear regression in Prism 7 (GraphPad Software).

**Nucleosome DUB assays**. Generation of nucleosomes mono-ubiquitinated at Lys119 on Histone 2A was achieved by mixing 21 nM hUbe1, 150 nM UbcH5b$^{3C/S}$, 40 nM Ring1B$^{1-159}$–Bmi1$^{1-109}$, 2 µM 5-IAF-labelled wild-type Ubiquitin and 400–800 nM recombinant nucleosomes in assay buffer (20 mM Tris-HCl pH 7.5, 50 mM NaCl, 5 mM ATP, 2 mM MgCl$_2$ and 2 mM DTT) for 4 h at 37 °C. The reaction was then quenched by incubation with 0.2 U purified Apyrase (New England Biolabs) for 10 min at 37 °C. Activity assays were performed in 10 µL reactions at 37 °C using the indicated Calypso$^{43-471}$–ASX$^{207-340}$ and BAP1$^{1-729}$–ASXL1$^{247-366}$ proteins (~1.5 µM and ~500 nM for each Calypso–ASX and BAP1–ASXL1 proteins, respectively) or wild-type Calypso$^{43-404}$–ASX$^{207-340}$ and BAP1$^{1-712}$–ASXL1$^{247-366}$ complexes (~0.7–1 µM), and between 200 and 400 nM H2AK119Ub nucleosomes. Reactions were terminated by the addition of SDS-PAGE loading buffer containing β-mercaptoethanol. Proteins were separated on 16% SDS-PAGE gels, and reactions were analysed on a Las-3000 (Fuji Film) imager using the 5-IAF signal of 5-IAF labelled wild-type Ubiquitin. Reactions containing No DUB and wild-type (i.e. WT) Calypso$^{43-471}$–ASX$^{207-340}$ or BAP1$^{1-729}$–ASXL1$^{247-366}$ complexes were performed in sextuplets, while samples containing the indicated Calypso$^{43-471}$–ASX$^{207-340}$ or BAP1$^{1-729}$–ASXL1$^{247-366}$ mutants were performed in triplicates. For each set of experiments, bands corresponding to H2AK119Ub nucleosomes were quantified using Image Studio Lite (LI-COR). Uncropped gels for Fig. 5b, c and Supplementary Fig. 6c are shown in Supplementary Figs. 8, 9 and 11. Replicate gels are also included in Supplementary Figs. 8 and 9.

**Pulldown assays**. To perform binding assays, His$_6$-fused wild type and mutants Calypso$^{43-404}$–ASX$^{207-340}$ complexes were first immobilised on Ni$^{2+}$-NTA resin. Pulldown experiments were conducted by mixing the resin-bound Calypso$^{43-404}$–ASX$^{207-340}$ variants with purified wild-type Ubiquitin in pulldown

buffer containing 20 mM HEPES pH 7.5, 100 mM NaCl, 1 mM DTT, and 0.2% (v/v) Tween-20. Reactions were set up in a final volume of 200 µL, and incubated for 1 h at 4 °C in rotation. The reaction mixtures were then centrifuged at 13,500 r. c.f. for 2 min at 4 °C, and the resins were washed three times with pulldown buffer prior to addition of reducing SDS-PAGE sample buffer. Samples were resolved on 16% SDS-PAGE gels, and visualised using Coomassie Blue staining (refer to Supplementary Fig. 11 for the uncropped gel).

**Electrophoretic mobility shift assays**. Recombinant unmodified nucleosomes (40–80 nM), reconstituted with the labelled Widom 601 220 bp DNA sequence, were incubated with increased concentrations of the indicated Calypso$^{43-404}$–ASX$^{207-340}$, Calypso$^{43-471}$–ASX$^{207-340}$, or BAP1$^{1-729}$–ASXL1$^{247-366}$ complexes in assay buffer containing 10 mM HEPES pH 7.5, 100 mM NaCl, 0.1 mM DTT, and 5% (v/v) glycerol. Concentrations of 0.5, 1, 2, 4, and 8 µM were used for each Calypso–ASX complex, while 0.1, 0.2, 0.5, 1, and 2 µM were employed for BAP1$^{1-729}$–ASXL1$^{247-366}$. Assays were incubated for 1 h at room temperature in a final volume of 10 µL. Samples were analysed using native PAGE gel electrophoresis on 4–12% polyacrylamide gels, which were run for 2–3 h at 125 V in 25 mM Tris-HCl and 20 mM glycine at 4 °C. The gels were pre-run in the same buffer for at least 1 h at 125 V and 4 °C prior to sample loading. Bands were visualised on an Odyssey FC Imaging system (LI-COR) at 700 nm with a 10 min exposure. Reactions containing wild type and mutant Calypso$^{43-471}$–ASX$^{207-340}$ and BAP1$^{1-729}$–ASXL1$^{247-366}$ complexes were performed in triplicates, and bands corresponding to the shifted nucleosomes were quantified using Image Studio Lite (LI-COR). The uncropped gel for Fig. 6a is shown in Supplementary Fig. 7, while replicate gels for Fig. 6b, c are provided in Supplementary Fig. 10.

**Figure generation**. All structural figures, including the electrostatic potential molecular surface of the nucleosome particle shown in Fig. 6e and the structure overlays shown in Supplementary Fig. 2, were created using PyMOL (The PyMOL Molecular Graphics System, Version 1.8.6.2 Schrödinger, LLC). Mapping of the BAP1 missense mutations onto the Calypso$^{43-404}$–ASX$^{207-340}$ structure shown in Fig. 2a was achieved using the loadBfacts.py script for PyMOL (https://pymolwiki.org/index.php/Load_new_B-factors). Electron density maps shown in Supplementary Fig. 5c were calculated using the Fast Fourier Transform (FFT) algorithm[67,68] within the CCP4 program suite[69]. Composite omit maps shown in Supplementary Fig. 5b, d were calculated with PHE-NIX[52] and FFT[67–69]. All maps were then overlaid onto the cartoon representation of the Calypso$^{43-404}$–ASX$^{207-340}$ structure in PyMOL. Analysis of the interaction surfaces in the Calypso$^{43-404}$–ASX$^{207-340}$ complex (see Supplementary Table 4) was performed using the European Bioinformatics Institute (EBI) web service of the Proteins, Interfaces, Structures and Assemblies (PISA) software (http://www.ebi.ac.uk/pdbe/pisa/)[38]. Graphs shown in Figs. 2c, 5, 6b, c and Supplementary Fig. 6b, as well as the representation of SAXS data shown in Fig. 4d and in Supplementary Fig. 5f were created using Prism 7 (GraphPad Software), while the elution profiles obtained from SEC-MALLS experiments (see Figs. 3a and 4c) were generated using Plot2 (Plot2, Version 2.0.8; http://plot.micw.eu/). AUC plots shown in Fig. 3b, c and in Supplementary Fig. 4 were generated using Origin (OriginLab, Northampton, MA). Sequence alignments shown in Fig. 2b, Supplementary Figs. 1b–d and 6a were created using Aline, Version 1.0.025 (ref. [70]). All figure panels were assembled using Adobe Illustrator CS4, Version 14.0.0 (Adobe Systems).

**Quantification and statistical analysis**. Statistics for X-ray data collection and refinement are summarised in Supplementary Table 1, values obtained by PISA for the analysis of the crystallographic interfaces in the Calypso$^{43-404}$–ASX$^{207-340}$ structure are reported in Supplementary Table 4, and statistics for SAXS data collection and analysis are summarised in Supplementary Table 5. Experiments shown in Figs. 2c, 5a, and in Supplementary Fig. 6b were performed twice, with error bars either indicating the range of measurement (Fig. 2c) or ±SEM (Fig. 5a and Supplementary Fig. 6b). Experiments shown in Fig. 5b, c were performed in sextuplets (for the No DUB and WT reactions) or triplicates (for reactions containing Calypso–ASX and BAP1–ASXL1 mutant complexes) while experiments shown in Fig. 6b, c were performed in triplicates, and the values corresponding to the six or three independent experiments with a line passing through the mean are shown.

## Data availability

The atomic coordinates and structure factors for the Calypso$^{43-404}$–ASX$^{207-340}$ complex have been deposited to the Protein Data Bank under ID code 6CGA. All data supporting the findings in this study are available within the article and in the Supplementary Information Files, and are available from the corresponding author upon reasonable request.

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

## Acknowledgements

We thank Sam Jamieson for technical assistance with various aspects of the project, Adam Graham for preliminary construct testing, Alan Riboldi-Tunnicliffe for diffraction screening at the Australian Synchrotron, and Michael Algie for helpful discussions. P.D.M. was supported by a Rutherford Discovery Fellowship from the New Zealand government administered by the Royal Society of New Zealand. Support was provided by a Project Grant from the Health Research Council of New Zealand, the Biochemistry Department at the University of Otago, a University of Otago Research Grant (to P.D.M.), the Victorian State Government Operational Infrastructure Support, NHMRC IRIISS grant (9000433), and an NHMRC fellowship (1105754; to J.M.M.). This research was undertaken in part using the small-angle X-ray scattering and the MX2 beamline at the Australian Synchrotron, part of ANSTO, and made use of the ACRF detector. We thank the New Zealand synchrotron group for facilitating access to the MX beamlines, and the Biomolecular Interaction Centre for AUC access.

## Author contributions

Conceptualisation: P.D.M.; methodology: P.D.M. and M.F.; formal analysis: M.F., A.J.M., J.M.C., R.C.J.D., J.M.M. and P.D.M.; investigation: M.F., A.J.M., A.E.B. and J.M.C.; resources: P.D.M., C.L.D., R.C.J.D., and J.M.M.; writing—original draft: P.D.M.; writing—review & editing: P.D.M., M.F., C.L.D. J.M.M. and R.C.J.D.; visualisation: M.F. and P.D.M.; supervision: P.D.M., C.L.D. and R.C.J.D.; project administration: P.D.M; funding acquisition: P.D.M., C.L.D., J.M.M. and R.C.J.D.

## Additional information

**Competing interests:** The authors declare no competing interests.

