## [Peer Review File · Nature Communications]

Reviewers' comments:

Reviewer #1 (Remarks to the Author):

The manuscript by Foglizzo et al. describes the first crystal structure of a PR-DUB complex, a histone deubiquitinase that counters the H2AK119Ub1 mark conferred by PRC1. The authors purify and crystallize the Calypso/ASX complex, which comprise the *Drosophila* homologs to the human deubiquitinase BAP1 and activating binding partner, ASXL1. In human cancers, BAP1/ASXL1 are frequently mutated and are associated with poor disease outcomes. Accordingly, the Calypso/ASX structure provides a valuable framework on which to study the molecular consequences of disease-associated BAP1/ASXL1 mutations. The authors also reveal that Calypso/ASX form a 2:2 complex, and identify Leu340 in Calypso as a primary mediator of oligomerization. They mutate this residue to alanine to generate a 1:1 Calypso/ASX, and show that although the 1:1 entity retains inherent deubiquitinase activity, the full 2:2 complex is essential for activity on H2AK119Ub nucleosomes. Lastly, they propose a revised model for PR-DUB regulation, in which the complex exists as a constitutive 1:1 complex with low deubiquitinase activity, and is enriched at specific genomic landmarks to induce 2:2 oligomerization and, in turn, recruitment of nucleosomal substrates to the bidentate C-terminal extensions of Calypso for deubiquitination.

The work appears to be carefully done, and the data and figures are well organized. One issue of note is that the previous work described by Sahtoe et al. (*Nat Comm*, 2016) is quite understated. Although not a formal PR-DUB, UCH-L5/Rpn13 is a closely related deubiquitinating complex with similar architecture and has previously served as a surrogate system for understanding PR-DUB biology. Figure S2 testifies to the high structural similarity of the Calypso/ASX and UCH-L5/Rpn13 complexes, and a discussion of the relevance of this earlier work would be appropriate in the Introduction.

The text would also be improved by a discussion of how Calypso residues Leu340 and Met288 were identified as primary mediators of Calypso/ASX oligomerization. Although the importance of Leu340 is established through biochemical and biophysical assessments, Met288 is not discussed. Presumably a M288A mutation (or equivalent) was generated – if so, with what outcome? Given that these residues were identified in the PISA analysis of Interface 5 (Table S4) – what are their specific contributions to the interface (e.g. hydrophobic pockets, backbone contacts, polar interactions, etc)? Are these the only residues involved? If appropriate, the addition of distance measurements for Figs. 4A and S5B would improve the clarity of the figures. Lastly, Fig S5B appears to lack electron density for the terminal thioether in Met288. If this is the case, it seems a bit of a reach to claim the residue as "involved in dimerisation" if the specific rotameric identity is unknown. Please also specify the contour level for the electron density.

In summary, the manuscript is well written and provides a convincing model for how PR-DUB complexes are organized and regulated. The structural and biochemical data are robust, and suggest that the clustering patterns of PR-DUB complexes dictate unique outcomes for H2AK119Ub. As such, the findings are of interest and significance to a number of related fields. Overall, we recommend this manuscript for publication in *Nature Communications* upon acknowledgment of our comments and concerns.

Specific comments

1. Figure 2B – Consider labeling the residues by color to distinguish ubiquitin/Calypso/ASX chains.
2. Figure 4C – It is not apparent why the 2:2 complex does not run as a ~110 kDa Calypso/ASX complex, given that the 1:1 complex prominently features at 55 kDa. Please label the identity of the high molecular weight species. Would a native gel provide adequate resolution of the various complexes?

3. Figure S3A – Electron density around Arg288 appears to be lacking. Please label the residues by color for clarity, and provide the contour level for the density. Is it the 2fofc density?
4. Figure S5A – Please provide an overlay of molecular weight standards for the gel filtration column for comparison. L340A and N243A appear to form high molecular weight species – are these aggregates, or higher order oligomers? Similar to earlier, a rationale for the selection of these residues (presumably from a PISA inquiry) should be provided in the Supplemental section. Also, “Interface #5” is colored red – what is the significance of this? Are all the mutants located at this interface?
5. The text on page 6 claims that the Calypso/ASX structure proves to be an “excellent” model for understanding BAP1/ASXL1/2 mutations. We advise using “promising” as the descriptor, as the current model lacks the large central insertion of BAP1 and the majority of findings are highly specific to the Drosophila complex.
6. Based on the crystal structure, they were able to map a few cancer-related mutations in ASXL1 or ASXL2 (Glu284, Arg 288) to ASX. These mutations seem to be loss-of-function mutations, but it is unclear whether they inactivate the complex by disrupting binding between ASX and Calypso, or by inactivating the enzymatic activity of the complex. Therefore, they should test whether these ASX mutants still complex with Calypso, or not.
7. They identified L340 in Calypso as a critical residue to mediate the dimerization between the PR-DUB complexes, but this residue appears to locate in a region distinct from the ASX binding interface. A reasonable hypothesis would be that ASX should not be required for such dimerization. Therefore, they should test whether Calypso alone can form dimers.
8. Other than L340, the dimerization interface contains multiple residues that could be required. The authors should test whether mutating other residues (such as M288) also disrupts dimerization. This could rule out the possibility that L340A disrupts local protein folding.
9. The authors also mutated the corresponding L635 in human BAP1 and showed that it also decreased its activity towards nucleosomes. As a control, they should test this mutation in their ubiquitin-AMC assay to confirm it is specific to nucleosomes as they speculate.
10. In fig. 6b, the authors claimed dramatic decrease of nucleosome shifting using the L340A mutant, but the changes look subtle. The authors should provide quantification for this assay. Additionally, per their hypothesis, if the dimerization promotes nucleosome binding, would they observe an increased shift using cross-linked Calypso-ASX complex?
11. A straightforward step following this hypothesis would be to express the L635A BAP1 mutant in cells and evaluate if it really results in a loss of function PR-DUB complex.
12. In fig.7, the authors proposed a model to explain how this dimerization could promote PR-DUB recruitment and activity toward nucleosomes. However, it doesn't explain why truncating mutations identified in ASXL1 often lead to gain of function and hyper-deubiquitinating activity. They should experimentally compare full-length ASXL and the truncated ASXL mutants in their assays, especially as the prior studies seem to suggest the opposite.

Reviewer #2 (Remarks to the Author):

This manuscript by Foglizzo et al. is focused on understanding the mechanism by which PR-DUB deubiquitinates H2AK119Ub. The authors have determined the crystal structure of Drosophila PR-DUB (comprising a complex of Calypso and a fragment of Asx that harbors the ULD domain) and performed

a nice combination of biochemical and biophysical experiments to probe their structural findings. Based on their studies, the authors propose a model for regulation of PR-DUB activity involving dimerization/oligomerization. The manuscript is clearly presented, the underlying data are of high quality, and the key findings potentially make a significant contribution to our understanding of how PR-DUB is regulated for removal of ubiquitin from Lys119 of histones and the role that mutations play in pathological states. With that said, there are important issues that need to be addressed in order to further strengthen the author's major conclusions.

Major points:

*Supplementary Table 1- The completeness reported in the table (98.8%) is significantly different from that listed in the PDB validation report (83%). The authors should clarify this discrepancy.

*Figure 5- related to 2:2 vs 2:1 stoichiometry. Can the authors perform the assays presented in Fig. 5 b,d with Calypso alone and then titrate Asx into the reactions and perhaps do this for both the human and drosophila complexes to see if there is an apparent difference in stoichiometry vs. activity for the complexes?

*Page 6&7- The molecular determinants of the observed 2:2 complex are not clear. Is the presence of ASX required for the dimerization of Calypso suggested by SEC-MALLS and AUC experiments performed on the Calypso/Asx complex (and if so is it required at a 2:1 or 2:2 ratio)? Can the Calypso ULD be expressed alone and does it exhibit the same tendency to dimerize?

*Given that the reported stoichiometry of human Bap1/ASXL1 complex is reported to be 2:1, what is the role of ASXL1 in regulating the function of the presumed Bap1 dimer in the human complex? Does one molecule of ASXL1 activate both protomers of Bap1 in the complex or is only one activated Bap1 protomer required for activity in the human system? Overall, this potential difference in stoichiometry between the human and Drosophila complexes is something the authors should investigate in more detail, given the importance of oligomeric state to their model. Relatedly, Figure 7 seems to indicate a 2:2 human complex stoichiometry, which has not been demonstrated.

*The authors have not convincingly demonstrated that the L340A mutation disrupts the 2:2 stoichiometry of the Calypso/Asx complex. L340A mutant Calypso should be subjected to AUC and/or SEC-MALLS since that is the method that was used to conclude that the stoichiometry of the wild type Calypso/Asx complex is 2:2. The observed shift in migration on the gel filtration column could be due to other factors such as altered conformation and also, there is no standard curve to assess the meaning of the shift (presumably due to the nonglobular nature of the complex)

*The effect of L340A is striking but the contacts this residue is involved in at the interface is not clear. Met288 is shown in figures but are there other residues that might have side chains involved in contacts to L340A? Were any other mutations in proximity to L340 evaluated, and did they have such a striking effect? Are there substitutions in proximity to L340 that could be made to compensate for the apparently deleterious effects of the L340A substitution to 2:2 complex formation?

*Page 7- How reliable is modeling the disordered loop of Calypso as a means of fitting the model to the SEC-SAXS data, given that the loop likely adopts a range of conformations? How is the fit and what is the Chi value for the Calypso/Asx complex without the loop modeled, and how does this all compare to the other binding modes? (I presume the data that was not shown was for other binding modes, not stoichiometries as was written).

*Page 8- A 2:2 Calypso/Asx complex is ~110 kDa but based on the data in Fig. 4c the crosslinks

appear to migrate between 150-250 kDa. Can the authors repeat the experiment with Calypso only (or titrating Asx) to potentially make interpretation more straightforward?

*Page9- Are two basic tails necessary for PR-DUB activity or only one? I ask this question with the 2:1 human complex in mind, in the context of structural data from of UCH-L5 complexes indicating conformational changes upon RPN13/INO80G binding. This could be addressed with a 'no tail' Calypso titration assay.

*Page9- Have the authors performed the nucleosome assays presented in Fig. 5b,d with the 'no tail' construct?

Minor points:

*Did the authors examine crystal packing in other related structures, such as the UCH-L5 complexes to see if there is a related 2:2 complex in the crystals? Is the residue corresponding to L340 of Calypso conserved?

*Fig 1c. and other figures should be labeled to make it clear that the ubiquitin molecule is modeled

*Page 5- What do the three helices observed in the previously reported Rpn13 structure that are not visualized in the Calypso/ASX structure reported here do? Are they involved in contacts to UCH-L5? Is it possible to melt Calypso/Asx crystals and either subject them to mass spectrometry or SDS-PAGE to determine whether residues 310-340 are in the crystal?

*Page 5- what is the crossover-loop

*Page 6- The authors show that E284 and R288 of Asx are predicted to be proximal to ubiquitin based on their model, and that mutations found in cancer patients compromise Calypso/Asx ability to hydrolyze ubiquitin-AMC or model ubiquitin-peptide substrates. It seems that the implication is that the mutations affect the ability of Calypso/Asx to interact with ubiquitin, but this is not demonstrated.

Page 6- define abbreviations such as AUC and SEC-MALLS.

Reviewer #3 (Remarks to the Author):

Foglizzo et al. describe the crystal structure of fly PR-DUB complex, which is an important enzyme that removes mono-ubiquitin from H2A. The central theme of the paper is the discovery of a higher order complex responsible for deubiquitylating H2A-K119 ubiquitin. The discovery of this bidentate complex, backed up by mutagenesis and enzyme activity data, provides the mechanistic basis for how PR-DUB activity can be regulated at specific genomic foci. This is an important study that provides important insights on the functions of PR-DUB activity and regulation that are relevant to understanding the biology of normal and malignant cells.

The study will be of interest to the readers of Nature Communication as well as others who are interested in gene expression, chromatin regulation, ubiquitin signalling and tumorigenesis. As such, this reviewer recommends publication in Nature Communication, provided that the authors address the following minor comments.

1. In figure 5A the authors show that WT and L340A mutants have similar activities against Ub-AMC. They conclude that this shows the bidentate complex is not important for cleaving mono-Ub which is not attached to a protein substrate. However, the enzyme assays are performed at 0.5 nM enzyme,

which is far lower than the low micromolar concentration where the bidentate complex is formed. At low nanomolar concentrations, the WT and L340A mutants will not be expected to be different anyway, because there is no bidentate complex.

The authors need to repeat this experiment at low micromolar concentration if possible, although specific activity may be too high to perform a Michaelis-Menten analysis. If they cannot do this, the authors need to be careful how they interpret the current results (Fig. 5A). The current data instead serve as a good check that the L340A mutant does not cause any unexpected structural disruption and helps in interpreting the assays with ubiquitylated histones in subsequent panels.

2. Supp. Fig. 3a. The authors need to mention the which type of electron density they are showing and at what sigma level. There is no density around Arg288, which contradicts the way the figure legend is written.

3. Fig. 4c and Supp. Figure 5e. The authors mention three separate bands (above 50 kDa) after crosslinking. This is not clear in the figure as only two bands are obvious.

General comment on additional data added in revision

Reviewer 1 (and Reviewer 2) both point out reliance on the Leu340 mutant protein for identifying the dimerisation interface. In the revised manuscript we have mutated both Met288 and Asn292 of Calypso at the same interface, coexpressed with ASX and purified, and used SEC-MALLS and analytical size-exclusion chromatography to demonstrate that both mutations disrupt the 2:2 complex. The same mutants showed decreased activity on H2AK119Ub nucleosomes. We also generated a mutant in the BAP1 residue corresponding to Calypso N292 (i.e. N251R), which after co-expression with ASXL1 in insect cells displays reduced activity on nucleosomes. We have incorporated this mutagenesis data to Fig. 4a-c (analytical SEC and SEC-MALLS) and Fig 5b,c (for activity data), and revised the related text on pages 7 and 8.

In addition, we have performed nucleosome binding experiments with wild-type and L635A BAP1–ASXL1 complexes (refer to Fig. 6c). Our data shows that the wild-type BAP1–ASXL1 complex shifts the parent nucleosome band much more efficiently than the BAP1(L635A)–ASXL1 protein. This confirmed that higher-order oligomerisation of both *Drosophila* and *Human* PR-DUBs is required for efficient nucleosome recruitment.

Several other specific pieces of data have also been added in response to reviewer comments, and are described alongside each relevant query.

Reviewers' comments:

Reviewer #1 :

In summary, the manuscript is well written and provides a convincing model for how PR-DUB complexes are organized and regulated. The structural and biochemical data are robust, and suggest that the clustering patterns of PR-DUB complexes dictate unique outcomes for H2AK119Ub. As such, the findings are of interest and significance to a number of related fields. Overall, we recommend this manuscript for publication in Nature Communications upon acknowledgment of our comments and concerns.

We thank the reviewer for their positive comments, and specific suggestions which have greatly added to the manuscript.

- *One issue of note is that the previous work described by Sahtoe et al. (Nat Comm, 2016) is quite understated. Although not a formal PR-DUB, UCH-L5/Rpn13 is a closely related deubiquitinating complex with similar architecture and has previously served as a surrogate system for understanding PR-DUB biology. Figure S2 testifies to the high structural similarity of the Calypso/ASX and UCH-L5/Rpn13 complexes, and a discussion of the relevance of this earlier work would be appropriate in the Introduction.*

In light of the comment from the reviewer, previous findings related to the UCH-L5–Rpn13 complex have been incorporated in the Introduction section of the revised text as shown below.

Page 4: “No structures of any PR-DUB complex have been reported, but inferences have been extended from the human ortholog UCH-L5, which also contains an N-terminal UCH domain and a C-terminal ULD domain. Recent studies have shown that substrate binding and catalysis by UCH-L5 is promoted by the Deubad of the proteasome-associated adapter Rpn13, or inhibited by the Deubad of the chromatin remodelling complex subunit INO80G, with the Deubad from each binding partner stabilising different conformations of the UCH-L5 domain^{18,19}. While the UCH-L5–Rpn13 complex has provided a template for PR-DUB catalysis and modelling of some cancer-derived mutations^{8,9,20}, many outstanding questions remain unanswered.”

- *The text would also be improved by a discussion of how Calypso residues Leu340 and Met288 were identified as primary mediators of Calypso/ASX oligomerization. Although the importance of Leu340 is established through biochemical and biophysical assessments, Met288 is not discussed. Presumably a M288A mutation (or equivalent) was generated – if so, with what outcome? Given that these residues were identified in the PISA analysis of Interface 5 (Table S4) – what are their specific contributions to the interface (e.g. hydrophobic pockets, backbone contacts, polar interactions, etc)? Are these the only residues involved?*

Please see the additional data comment at top of rebuttal.

- *If appropriate, the addition of distance measurements for Figs. 4A and S5B would improve the clarity of the figures.*

Thanks for the suggestion. We have incorporated distance measurements into Supplementary Fig. 5b, but left Figure 4c (previously 4a) without, to avoid adding confusion.

- *Lastly, Fig S5B appears to lack electron density for the terminal thioether in Met288. If this is the case, it seems a bit of a reach to claim the residue as “involved in dimerisation” if the specific rotameric identity is unknown. Please also specify the contour level for the electron density.*

The reviewer has raised a fair point regarding a lack of electron density for the terminal thioether in Met288, presumably owing to the overall resolution and diffraction data anisotropy. However, our biophysical and biochemical experiments (refer to Fig. 4a-c) have confirmed that mutation of this residue does indeed disrupt formation of the 2:2 PR-DUB complex, and hence we judge it is still accurate to say that Met288 is involved. We have included the contour level for the electron density in the revised manuscript.

Specific comments

1. *Figure 2B – Consider labeling the residues by color to distinguish ubiquitin/Calypso/ASX chains.*

We have now labeled residues in Fig. 2b with the same color as the corresponding molecules.

2. *Figure 4C – It is not apparent why the 2:2 complex does not run as a ~110 kDa Calypso/ASX complex, given that the 1:1 complex prominently features at 55 kDa. Please label the identity of the high molecular weight species. Would a native gel provide adequate resolution of the various complexes?*

This is a fair point. We did consider it unusual at first that crosslinked species run at a higher apparent molecular weight than calculated, but we concluded this is due to aberrant behaviour on SDS-PAGE. Our SEC-MALLS and AUC data, which are more quantitative, clearly support formation of a 2:2 complex. While native gels are one option we feel our new SEC-MALLS experiments add quite strong orthogonal data, but we have modified the text to note the aberrant mobility as shown below.

Pages 7 and 8: “A second tight group of crosslinked bands migrated with a molecular weight of approximately 150 kDa. [...] Although SDS-PAGE mobility was slightly aberrant relative to a theoretical mass of 109 kDa for the 2:2 complex, the simplest explanation for this profile is that the high molecular weight species present contain Calypso-Calypso crosslinks. These crosslinks are specific to wild-type Calypso-ASX rather than the L340A protein, and consistent with L340A disrupting the Calypso-Calypso coiled-coil interface, but not disrupting the ability of Calypso to bind ASX.”

3. *Figure S3A – Electron density around Arg288 appears to be lacking. Please label the residues by color for clarity, and provide the contour level for the density. Is it the 2fofc density?*

We have labeled residues in Supplementary Fig. 3a by color, and have provided the contour level for the 2Fo-Fc electron density map in the revised figure legend.

4. *Figure S5A – Please provide an overlay of molecular weight standards for the gel filtration column for comparison. L340A and N243A appear to form high molecular weight species – are these aggregates, or higher order oligomers? Similar to earlier, a rationale for the selection of these residues (presumably from a PISA inquiry) should be provided in the Supplemental section. Also, “Interface #5” is colored red – what is the significance of this? Are all the mutants located at this interface?*

We have included the elution profiles of molecular weight standards in the revised manuscript (see Fig. 4a,b, *top arrows*). As pointed out by the reviewer, Calypso–ASX(N243A) elutes marginally prior to WT protein, and L340A exhibits a small peak in a similar position. We can confirm that the earlier peak of L340A contains the complex as determined by SDS-PAGE, but runs well clear of the void volume. While we are unsure about the details of such behaviour, this was not consistently observed (for instance see SEC-MALLS data in new Fig. 4c, which was loaded at similar protein concentration). To help identify the position of the different mutants Supplementary Fig. 5a has now been included.

5. *The text on page 6 claims that the Calypso/ASX structure proves to be an “excellent” model for understanding BAP1/ASXL1/2 mutations. We advise using “promising” as the descriptor, as the current model lacks the large central insertion of BAP1 and the majority of findings are highly specific to the Drosophila complex.*

This is a point well taken, and the text has been modified as suggested.

6. *Based on the crystal structure, they were able to map a few cancer-related mutations in ASXL1 or ASXL2 (Glu284, Arg 288) to ASX. These mutations seem to be loss-of-function mutations, but it is unclear whether they inactivate the complex by disrupting binding between ASX and Calypso, or by inactivating the enzymatic activity of the complex. Therefore, they should test whether these ASX mutants still complex with Calypso, or not.*

The Calypso–ASX complexes used in this study have all been co-expressed and co-purified as this greatly improved protein behaviour. Therefore, we are confident that the ASX mutants E284K, E284Q and R288N still bind Calypso and form a stable complex. However, as hypothesised by the reviewer, these same mutants inhibit enzymatic activity of the complex due to reduced ubiquitin binding (refer to Fig. 2b,c, and Supplementary Fig. 3b,c). This aspect has been clarified in the revised text as shown below.

Page 5: “[...] Consistent with the Calypso–ASX structure, all mutant complexes lost the ability to hydrolyse Ubiquitin-AMC or a model ubiquitin-peptide substrate, and had reduced capacity to bind ubiquitin in pulldown assays (Fig. 2c and Supplementary Fig. 3b,c).”

7. *They identified L340 in Calypso as a critical residue to mediate the dimerization between the PR-DUB complexes, but this residue appears to locate in a region distinct from the ASX binding interface. A reasonable hypothesis would be that ASX should not be required for such dimerization. Therefore, they should test whether Calypso alone can form dimers.*

This is a good point raised by the reviewer. It is still possible that such dimerisation is triggered by ASX binding to the Calypso ULD. In light of the comment we have since re-attempted expression of both wild-type and L340A Calypso proteins without ASX, in two insect cell lines, but we have found them to be largely insoluble and not tractable for characterisation. We thank the reviewer for their suggestion.

8. *Other than L340, the dimerization interface contains multiple residues that could be required. The authors should test whether mutating other residues (such as M288)*

also disrupts dimerization. This could rule out the possibility that L340A disrupts local protein folding.

Please see the additional data comment at top of rebuttal

- 9. The authors also mutated the corresponding L635 in human BAP1 and showed that it also decreased its activity towards nucleosomes. As a control, they should test this mutation in their ubiquitin-AMC assay to confirm it is specific to nucleosomes as they speculate.*

As suggested by the reviewer, we have performed Ubiquitin-AMC assays with both wild-type and L635A BAP1-ASXL1 proteins. Similar to what we have previously demonstrated for Calypso-ASX, our data shows that both complexes have comparable activities when tested on a model substrate. This indicates that the difference in activity between wild-type and mutant BAP1-ASXL1 proteins is specific to nucleosomes as shown for the corresponding *Drosophila* complexes. We have included these new data in Supplementary Fig. 6b of the current submission, and have revised the current version of the manuscript as shown below.

Pages 9/10: "Wild-type and mutant BAP1-ASXL1 complexes were first compared in Ubiquitin-AMC cleavage assays. As previously shown for the corresponding Calypso-ASX proteins, we did not detect a noticeable difference in K_M between the two BAP1-ASXL1 constructs (Supplementary Fig. 6b), demonstrating that mutation at Leu635 did not induce structural changes in BAP1 that could affect its catalytic activity."

- 10. In fig. 6b, the authors claimed dramatic decrease of nucleosome shifting using the L340A mutant, but the changes look subtle. The authors should provide quantification for this assay. Additionally, per their hypothesis, if the dimerization promotes nucleosome binding, would they observe an increased shift using cross-linked Calypso-ASX complex?*

We have now strengthened this section by including quantification of the assays (Revised Fig. 6b). In addition, we have performed nucleosome binding experiments with wild-type and L635A BAP1-ASXL1 complexes. Our data shows that the wild-type BAP1-ASXL1 complex shifts the parent nucleosome band much more efficiently than the BAP1(L635A)-ASXL1 protein. This confirmed that higher-order oligomerisation of both *Drosophila* and *Human* PR-DUBs is required for efficient nucleosome recruitment. We have revised the text in light of this new data as shown below.

Pages 10/11: "[...] Performing equivalent experiments with the corresponding BAP1-ASXL1 complexes demonstrated that wild-type BAP1-ASXL1 readily shifted the nucleosome band, whereas the L635A BAP1-ASXL1 complex had a dramatically impaired ability to induce nucleosome mobility shift at the same concentrations (Fig. 6c)."

- 11. A straightforward step following this hypothesis would be to express the L635A BAP1 mutant in cells and evaluate if it really results in a loss of function PR-DUB complex.*
- 12. In fig.7, the authors proposed a model to explain how this dimerization could promote PR-DUB recruitment and activity toward nucleosomes. However, it doesn't explain why truncating mutations identified in ASXL1 often lead to gain of function and hyper-deubiquitinating activity. They should experimentally compare full-length ASXL and the truncated ASXL mutants in their assays, especially as the prior studies seem to suggest the opposite.*

Points 11 and 12 are indeed excellent points that are the subject of our ongoing studies. With regard to gain-of-function due to truncation of ASXL1/2, we understand this question is topical in the field—our model would predict that truncating mutations may have moderate basal activity, but untethered to particular foci (for instance by PHD-based targeting), hence globally erasing H2AK119Ub. In effect the mechanism we have come upon could be equally important for promoting on-target activity, and restricting complexes away from untargeted regions.

When we transfect WT and BAP1 L635A/E in HEK293 cells there are marked differences in stability, which are most obvious in the context of truncated ASXL1. This suggests intriguing different behaviours of WT and oligomer deficient complexes, but to disentangle the effects of stability, activity, and localisation (e.g. Mashtalir *et al.*, 2014) will require dedicated studies that we are developing. Nonetheless, the different behaviour of WT and mutant BAP1 reiterates the importance of higher order oligomerisation, as we alluded to by raising auto-deubiquitination in the discussion.

Reviewer #2 (Remarks to the Author):

This manuscript by Foglizzo et al. is focused on understanding the mechanism by which PR-DUB deubiquitinates H2AK119Ub.The manuscript is clearly presented, the underlying data are of high quality, and the key findings potentially make a significant contribution to our understanding of how PR-DUB is regulated for removal of ubiquitin from Lys119 of histones and the role that mutations play in pathological states. With that said, there are important issues that need to be addressed in order to further strengthen the author's major conclusions.

We thank the reviewer for their positive comments.

Major points:

- *Supplementary Table 1- The completeness reported in the table (98.8%) is significantly different from that listed in the PDB validation report (83%). The authors should clarify this discrepancy.*

We apologise for this discrepancy, which arose because of the inherent anisotropy in our diffraction data. During refinement we noticed improved maps after application of anisotropy correction (Strong *et al*, *PNAS* 2006), with data along the *b* and *c* axis clearly extending to ~3.4/3.5 Å, but the *a* axis being significantly poorer. We noted use of anisotropy corrected data in the methods, but there doesn't appear to be a clear consensus about how to report this in the literature. We have since performed re-refinement with the full dataset (98.6% complete to 3.5 Å), which has marginally worse R-factors (26/30.2) but no noticeable changes in the model. Furthermore, we have discussed with scientists who run the diffraction anisotropy server at UCLA, who suggested we could report data statistics both with and without anisotropic scaling.

We propose to include an additional line in the crystallographic table illustrating refinement statistics with both full and anisotropy corrected data—but are open to further suggestions for best practise in this regard? We will also amend our PDB submission to include both datasets if possible.

- *Figure 5- related to 2:2 vs 2:1 stoichiometry. Can the authors perform the assays presented in Fig. 5 b,d with Calypso alone and then titrate Asx into the reactions and perhaps do this for both the human and drosophila complexes to see if there is an apparent difference in stoichiometry vs. activity for the complexes?*
- *Page 6&7- The molecular determinants of the observed 2:2 complex are not clear. Is the presence of ASX required for the dimerization of Calypso suggested by SEC-MALLS and AUC experiments performed on the Calypso/Asx complex (and if so is it required at a 2:1 or 2:2 ratio)? Can the Calypso ULD be expressed alone and does it exhibit the same tendency to dimerize?*
- *Given that the reported stoichiometry of human Bap1/ASXL1 complex is reported to be 2:1, what is the role of ASXL1 in regulating the function of the presumed Bap1 dimer in the human complex? Does one molecule of ASXL1 activate both protomers of Bap1 in the complex or is only one activated Bap1 protomer required for activity in the human system? Overall, this potential difference in stoichiometry between the human and Drosophila complexes is something the authors should investigate in*

more detail, given the importance of oligomeric state to their model. Relatedly, Figure 7 seems to indicate a 2:2 human complex stoichiometry, which has not been demonstrated.

- Page9- Are two basic tails necessary for PR-DUB activity or only one? I ask this question with the 2:1 human complex in mind, in the context of structural data from of UCH-L5 complexes indicating conformational changes upon RPN13/INO80G binding. This could be addressed with a 'no tail' Calypso titration assay.

All of these points related to complex stoichiometry are good ones, which we have attempted to answer by expressing the isolated DUB proteins (i.e. wild-type and L340A Calypso, and L635A BAP1 proteins) in two different insect cell lines. Unfortunately, both Calypso proteins were found to be largely insoluble (see response to Reviewer 1). As for the corresponding BAP1 proteins, wild-type BAP1 was expressed alone, but the L635A protein was predominantly insoluble. Thus, the experiments could not be performed.

While our structure shows that dimerisation of Calypso occurs via the coiled-coil hairpin of the two Calypso molecules (with ASX binding at a different interface), it is possible that such dimerisation is triggered by ASX binding to the Calypso ULD. Sahtoe et al. demonstrated that wild-type BAP1 in isolation forms dimers and higher order oligomers in solution (Sahtoe *et al.*, 2016), and Calypso may well behave in a similar manner. While Sahtoe et al report a 2:1 complex based on a MALLS experiment, they do suggest in the Discussion that this ratio requires further analysis. It is unclear from closer examination of the related mass spectrometry study (Baymaz et al 2014) whether a 2:1 ratio relates to 2:1 BAP1-ASXL1 and 2:1 BAP1-ASXL2 complexes, which would equate to equal stoichiometry between BAP1 and ASXL components overall. We have addressed this point in the Discussion section of the revised text (see page 13), as shown below.

“[...] Several lines of evidence suggest that the human PR-DUB also forms higher-order oligomers, and a conserved mode of deubiquitinase oligomerisation is consistent with our finding that mutating the coiled-coil of BAP1 reduces activity on H2AK119Ub nucleosomes (Fig. 5b,c). In vitro experiments have previously shown that BAP1 oligomerises and BAP1-ASXL1 can form a complex with apparent 2:1 stoichiometry^{9,37}, while mass spectrometry and biochemical studies have shown that the PR-DUB can consist of BAP1-ASXL1 or BAP1-ASXL2 complexes^{8,37}. It remains to be seen whether the equimolar (2:2) DUB-ASX stoichiometry we observe is conserved in humans but obscured by redundancy of human PR-DUB components. There is also scope for the BAP1-specific insert to influence the makeup of the PR-DUB in humans, or its stoichiometry, relative to the simpler two-component system of *Drosophila*.”

- The authors have not convincingly demonstrated that the L340A mutation disrupts the 2:2 stoichiometry of the Calypso/Asx complex. L340A mutant Calypso should be subjected to AUC and/or SEC-MALLS since that is the method that was used to conclude that the stoichiometry of the wild type Calypso/Asx complex is 2:2. The observed shift in migration on the gel filtration column could be due to other factors such as altered conformation and also, there is no standard curve to assess the meaning of the shift (presumably due to the nonglobular nature of the complex)
- The effect of L340A is striking but the contacts this residue is involved in at the interface is not clear. Met288 is shown in figures but are there other residues that might have side chains involved in contacts to L340A? Were any other mutations in proximity to L340 evaluated, and did they have such a striking effect? Are there substitutions in proximity to L340 that could be made to compensate for the apparently deleterious effects of the L340A substitution to 2:2 complex formation?

Please see the additional data comment at top of rebuttal.

- Page 7- How reliable is modeling the disordered loop of Calypso as a means of fitting the model to the SEC-SAXS data, given that the loop likely adopts a range of

conformations? How is the fit and what is the Chi value for the Calypso/Asx complex without the loop modeled, and how does this all compare to the other binding modes? (I presume the data that was not shown was for other binding modes, not stoichiometries as was written).

This is a valid point raised by the reviewer, and we apologise for the unclear wording. The additional loops have a relatively modest effect on the fit to scattering data, but given they are certainly present in solution we judged they should be included in the model. We have added the Chi values for the three binding modes (with and without loops) in Supplementary Fig. 5a (*bottom row*), which show that interfaces #3 and #5 both gave reasonable fits to the scattering profiles obtained from SEC-SAXS. Because it is difficult to resolve these complexes based on the resolution of SAXS, we have based our conclusions on mutagenesis and analytical SEC/SEC-MALLS (Fig. 4a-c), where mutations at interface #5 disrupt the 2:2 oligomer. This was also supported by our cross-linking experiments (see Fig. 4e and Supplementary Fig. 5e).

We have now clarified that SEC-SAXS experiments gave clear stoichiometry but similar Chi values for two of the interfaces on Page 8.

Pg 8: “The arrangement mediated by hydrophobic interactions between Leu340 and Met288 from the two Calypso coiled-coil hairpins (Supplementary Fig. 5b) creates an unusual elongated 2:2 complex, which we investigated in solution using SEC-SAXS. Guinier analyses showed that the Calypso–ASX complex was monodisperse and free of aggregation, with a calculated D_{\max} of 125 Å (Fig. 4d, Supplementary Fig. 5c, and Supplementary Table 5). This was consistent with a 2:2 Calypso–ASX complex being present in solution, in line with SEC-MALLS (Fig. 3a). [...] Having generated a complete model (Supplementary Fig. 5d), scattering profiles were an excellent fit for the oligomers formed through either the Calypso UCH and Calypso coiled-coil hairpin (Supplementary Fig. 5a, *bottom row*; interfaces #3 and #5) whereas interface #4 led to higher Chi values and poor fits (Supplementary Fig. 5a, *bottom row*).”

- *Page 8- A 2:2 Calypso/Asx complex is ~110 kDa but based on the data in Fig. 4c the crosslinks appear to migrate between 150-250 kDa. Can the authors repeat the experiment with Calypso only (or titrating Asx) to potentially make interpretation more straightforward?*

The reviewer has raised a good point, however as described above we have been unable to express the isolated Calypso protein in a soluble form and we have therefore not been able to perform the suggested experiments. We feel the additional MALLS data strengthens interpretation, but we have also revised the text to note this aberrant mobility following crosslinking (refer to pages 7 and 8 - “*PR-DUB oligomerisation via the coiled-coil hairpin*” section) as shown below.

“Although SDS-PAGE mobility was slightly aberrant relative to a theoretical mass of 109 kDa for the 2:2 complex, the simplest explanation for this profile is that the high molecular weight species present contain Calypso-Calypso crosslinks. These crosslinks are specific to wild-type Calypso–ASX rather than the L340A protein, and consistent with L340A disrupting the Calypso-Calypso coiled-coil interface, but not disrupting the ability of Calypso to bind ASX.”

- *Page9- Have the authors performed the nucleosome assays presented in Fig. 5b,d with the 'no tail' construct?*

We have performed nucleosome assays with both Calypso–ASX and BAP1–ASXL1 ‘no tail’ complexes. Consistent with the importance of the Calypso/BAP1 C-terminal positively-charged tail in PR-DUB recruitment to nucleosomes (see Fig. 6a and Sahtoe *et al.*, 2016), we have found that both complexes have severely impaired ability to cleave ubiquitin from nucleosomes mono-ubiquitinated at H2AK119. We have included these new data in Supplementary Fig. 6c, and have revised the text as below.

Pg 10: "In addition, we tested wild-type Calypso^{no tail}-ASX and BAP1^{no tail}-ASXL1 complexes for their ability to cleave ubiquitin from nucleosomes mono-ubiquitinated at H2AK119. Consistent with our binding experiments, as well as with a previous study by Sahtoe et al.⁹, both complexes showed impaired activity (Supplementary Fig. 6c)."

Minor points:

- *Did the authors examine crystal packing in other related structures, such as the UCH-L5 complexes to see if there is a related 2:2 complex in the crystals? Is the residue corresponding to L340 of Calypso conserved?*

As suggested by the reviewer, we examined the crystal packing in the structures of UCH-L5 in complex with either RPN13 (4UEM) or INO80G (4UF5) as well as in the structure of the UCH-L5~Ub/RPN13 complex (4UEL). We have found no evidence in any of these structures for the presence of a related 2:2 complex, as it appears in Calypso-ASX. Furthermore, despite the residue corresponding to Calypso L340 being relatively conserved in UCH-L5 (i.e. Met256), there is no evidence for UCH-L5 M256 to participate in complex formation from any of these structures. In addition, the degree of conservation in the surrounding residues is relatively low (refer to Supplementary Fig. 1c), further suggesting that UCH-L5 oligomerisation is not be mediated via this interface.

- *Fig 1c. and other figures should be labeled to make it clear that the ubiquitin molecule is modeled*

The modelled ubiquitin has been labelled throughout to avoid confusion.

- *Page 5- What do the three helices observed in the previously reported Rpn13 structure that are not visualized in the Calypso/ASX structure reported here do? Are they involved in contacts to UCH-L5? Is it possible to melt Calypso/Asx crystals and either subject them to mass spectrometry or SDS-PAGE to determine whether residues 310-340 are in the crystal?*

As suggested by the reviewer, we performed SDS-PAGE analysis at the time on the Calypso-ASX crystals and did not observe any noticeable difference compared to the purified complex. We strongly suspect that residues 310-340 of the ASX Deubad are in the crystal but not resolved in our electron density map, but our attempts at mass spectrometry were inconclusive so we are limited by the resolution of SDS-PAGE. Hence we felt the need to mention the possibility of digestion in the text.

- *Page 5- what is the crossover-loop*

The definition of "crossover-loop" has been included on pages 5 and 6 - "Characteristics of missense mutations in the human PR-DUB" section of the revised text.

- *Page 6- The authors show that E284 and R288 of Asx are predicted to be proximal to ubiquitin based on their model, and that mutations found in cancer patients compromise Calypso/Asx ability to hydrolyze ubiquitin-AMC or model ubiquitin-peptide substrates. It seems that the implication is that the mutations affect the ability of Calypso/Asx to interact with ubiquitin, but this is not demonstrated.*

This is a good point, and to support our hypothesis we have performed pull-down assays with wild-type and mutant Calypso-ASX proteins. Our new data show that mutations at residues E284 and R288 of ASX reduce the ability of Calypso-ASX to bind ubiquitin (see response to Reviewer 1). We have included these new binding data in Supplementary Fig. 3c, and revised the text as below.

Pg 5/6:"Consistent with the Calypso-ASX structure, all mutant complexes lost the ability to hydrolyse Ubiquitin-AMC or a model ubiquitin-peptide substrate, and had reduced capacity to bind ubiquitin (Fig. 2c and Supplementary Fig. 3b,c)."

- *Page 6- define abbreviations such as AUC and SEC-MALLS.*

All abbreviations have been defined in the revised version of the manuscript.

Reviewer #3 (Remarks to the Author):

This is an important study that provides important insights on the functions of PR-DUB activity and regulation that are relevant to understanding the biology of normal and malignant cells.The study will be of interest to the readers of Nature Communication as well as others who are interested in gene expression, chromatin regulation, ubiquitin signalling and tumorigenesis. As such, this reviewer recommends publication in Nature Communication, provided that the authors address the following minor comments.

We thank the reviewer for their positive comments.

1. *In figure 5A the authors show that WT and L340A mutants have similar activities against Ub-AMC. They conclude that this shows the bidentate complex is not important for cleaving mono-Ub which is not attached to a protein substrate. However, the enzyme assays are performed at 0.5 nM enzyme, which is far lower than the low micromolar concentration where the bidentate complex is formed. At low nanomolar concentrations, the WT and L340A mutants will not be expected to be different anyway, because there is no bidentate complex.*

The authors need to repeat this experiment at low micromolar concentration if possible, although specific activity may be too high to perform a Michaelis-Menten analysis. If they cannot do this, the authors need to be careful how they interpret the current results (Fig. 5A). The current data instead serve as a good check that the L340A mutant does not cause any unexpected structural disruption and helps in interpreting the assays with ubiquitylated histones in subsequent panels.

This is an excellent point. We have tried to perform Ubiquitin-AMC assays with micromolar concentration of wild-type and L340A Calypso-ASX complexes, but as predicted by the Reviewer the activity of both DUBs is too high for an accurate analysis. The text has therefore been revised as below.

Pg 9/10: “ We then used Ubiquitin-AMC cleavage assays to determine the Michaelis-Menten constant for both wild-type and L340A proteins. There was no significant difference in K_M between the constructs (Fig. 5a), showing that mutation at Leu340 did not affect the correct folding of the complex or its catalytic activity.”

2. *Supp. Fig. 3a. The authors need to mention the which type of electron density they are showing and at what sigma level. There is no density around Arg288, which contradicts the way the figure legend is written.*

The figure legend for Supplementary Fig. 3a has been revised as suggested by the reviewer.

3. *Fig. 4c and Supp. Figure 5e. The authors mention three separate bands (above 50 kDa) after crosslinking. This is not clear in the figure as only two bands are obvious.*

This is a fair point, and we apologise for the confusion. At high resolution there are three bands that run very close to each other (i.e. a triplet) at high molecular weight in the crosslinked SDS-PAGE, but these are far from visible in the figure we presented. We have revised the text as below.

Pg 7/8: “The first and most prominent complex migrated at ~55 kDa, consistent with a covalent 1:1 Calypso-ASX complex. A second tight group of crosslinked bands migrated with a molecular weight of approximately 150 kDa. “

REVIEWERS' COMMENTS:

Reviewer #1 (Remarks to the Author):

All cancers have been satisfactorily addressed.

Reviewer #2 (Remarks to the Author):

The authors have done a nice job of addressing concerns from all three reviewers by designing and conducting new experiments to test some of the interpretations of their data. Not all of the issues could be investigated due to an inability to generate the necessary reagents, but through their initial efforts and their efforts during the revision, the authors' major conclusions have been strengthened and manuscript now represents a major step forward in our understanding of how PR-DUB complexes are organized and regulated.

With regards to anisotropy correction, I agree that adding refinement statistics for both the full and anisotropy corrected data is warranted. The authors note that during refinement they noticed improved maps after application of anisotropy correction. I recommend adding a panel in the supplementary material showing composite omit map electron density for the overall structures and dimer interface for both the full and anisotropy corrected data, and perhaps replacing the 2Fo-Fc electron density maps presented in the current version of the supplementary figures with composite omit map density.

Reviewer #3 (Remarks to the Author):

The authors have addressed my concerns and I recommend publication.

Reviewers' comments:

Reviewer #1 (Remarks to the Author):

All cancers have been satisfactorily addressed.

We thank the reviewer for their positive comments, and specific suggestions which have greatly added to the manuscript.

Reviewer #2 (Remarks to the Author):

The authors have done a nice job of addressing concerns from all three reviewers by designing and conducting new experiments to test some of the interpretations of their data. Not all of the issues could be investigated due to an inability to generate the necessary reagents, but through their initial efforts and their efforts during the revision, the authors' major conclusions have been strengthened and manuscript now represents a major step forward in our understanding of how PR-DUB complexes are organized and regulated.

With regards to anisotropy correction, I agree that adding refinement statistics for both the full and anisotropy corrected data is warranted. The authors note that during refinement they noticed improved maps after application of anisotropy correction. I recommend adding a panel in the supplementary material showing composite omit map electron density for the overall structures and dimer interface for both the full and anisotropy corrected data, and perhaps replacing the 2Fo-Fc electron density maps presented in the current version of the supplementary figures with composite omit map density.

We thank the reviewer for their positive comments. In response to the concerns raised, we have replaced the 2Fo-Fc electron density map shown in Supplementary Figure 5b with a composite omit map, and we have introduced a 2Fo-Fc electron density map of the overall structure in Supplementary Figure 5c. In parallel, we have also added two additional panels (Supplementary Figure 5d) showing regions in the structure that had improved electron density after anisotropy correction. A table that further highlights the improvement of map-model correlation after anisotropy correction has been introduced in Supplementary Figure 5e.

Reviewer #3 (Remarks to the Author):

The authors have addressed my concerns and I recommend publication.

We thank the reviewer for their positive comments, and valuable insights which have greatly improved the manuscript.